# A universally applicable method of calculating confidence bands for ice nucleation spectra derived from droplet freezing experiments

William D. Fahy[1,†], Cosma Rohilla Shalizi[2,3], Ryan C. Sullivan[1,*]

[1]Center for Atmospheric Particle Studies, Carnegie Mellon University, Pittsburgh, PA 15213, USA
[2]Department of Statistics and Data Science, Carnegie Mellon University, Pittsburgh, PA 15213, USA
[3]Santa Fe Institute, Santa Fe, NM 87501, USA
[†]Now at: Department of Chemistry, University of Toronto, Toronto, ON M5S 3H6, Canada

*Correspondence to*: Ryan C. Sullivan, (rsullivan@cmu.edu)

**Abstract.** A suite of generally applicable statistical methods based on empirical bootstrapping is presented for calculating uncertainty and testing the significance of quantitative differences in temperature and/or ice active site densities between ice nucleation temperature spectra derived from droplet freezing experiments. Such experiments are widely used to determine the heterogeneous ice nucleation properties and ice nucleation particle concentration spectra of different particle samples, as well as in studies of homogeneous freezing. Our methods avoid most of the assumptions and approximations inherent to existing approaches and when sufficiently large sample sizes are used (approximately >150 droplets and >= 1000 bootstrap samples in our system) can capture the full range of random variability and error in ice nucleation spectra. Applications include calculation of accurate confidence intervals and confidence bands, quantitative statistical testing of differences between observed freezing spectra, accurate subtraction of the background filtered water freezing signal, and calculation of a range of statistical parameters using data from a single droplet array freezing experiment if necessary. By providing additional statistical tools to the community, this work will improve the quality and accuracy of statistical tests and representations of uncertainty in future ice nucleation research and will allow quantitative comparisons of the ice nucleation ability of different particles and surfaces.

## 1 Introduction

Ice nucleation (IN) is a complex process with significant implications for cloud properties in atmospheric science (Gettelman et al., 2012; Mülmenstädt et al., 2015; Froyd et al., 2022). Heterogeneous ice nucleation, where a separate phase or substance assists the nucleation of ice above the homogeneous freezing limit, is particularly difficult to study as the length and time scales at play in nucleation are difficult to directly observe (Fletcher, 1969; Wang et al., 2016; Kiselev et al., 2017; Holden et al., 2019). Most researchers resort to macroscopic measurements of this nanoscale process by creating droplets containing suspensions of the ice active material and observing freezing events as time passes or temperature changes (Vali, 2014). The most common technique is a variation on the droplet-on-substrate apparatus, where droplets of known sizes are created by manual pipetting, condensation, or microfluidic means (Stan et al., 2009; Budke and Koop, 2015; Whale et al.,

2015; Chen et al., 2018; Polen et al., 2018; Reicher et al., 2018; Brubaker et al., 2020; Gute and Abbatt, 2020; Roy et al., 2021). These droplets are usually exposed to a negative temperature ramp and the freezing temperatures of each droplet are recorded to produce an ice nucleation rate or active site density spectrum as a function of temperature (here we use the term 'IN activity' as a general term to refer to any measured or derived variable which quantifies ice nucleation rate with respect to temperature). Other procedures can be used to test the effects of time and other variables on IN activity (Wright and Petters, 2013).

Because these experiments only indirectly measure IN activity, results can have high natural variability, even when measuring the same sample on the same instrument. This variability is inherent to ice nucleation. Using the combined singular-stochastic VS66 model most recently discussed in Vali (2014) and terminology proposed in Vali et al. (2015), ice nucleation activity (or rate) is an accumulation of many ice nucleation sites with variable critical temperatures dispersed randomly throughout a material. In turn, the material is distributed randomly throughout droplets which can have varying sizes, shapes, and environments. Therefore, a measured IN activity can be affected by heterogeneity in the distribution of ice active sites across a material, heterogeneity in the mass or surface area of material suspended in each droplet, differences between droplet sizes and environments, and variations in temperature between droplets (Polen et al., 2018). Even in a perfect experimental setup, the stochastic nature of nucleation causes variation in the measured temperature dependence of a material's IN activity using a singular model (Vali, 2014, 2019). Combined with the large variations in IN activity observed between different ice nucleating substances and particles, this inherent uncertainty creates difficulties in reliably assessing whether differences in observed IN spectra indicate a statistically significant difference in IN activity.

Experimental error is always present and must be accounted for and reported, usually in the form of a standard error or a confidence interval of the mean measurement recorded. In our experience, there is no widely implemented approach to reporting uncertainty in IN temperature spectra derived from freezing experiments. Instead, methods vary between groups, relying on different assumptions about the nature of ice nucleation experiments, the forms of distributions that the random variables involved take, and the quantification of the derived uncertainties. In the simplest case, standard deviations, errors, and/or confidence intervals have been calculated from repeated experiments either by assuming that variability follows a normal distribution (Losey et al., 2018; Polen et al., 2018; Jahn et al., 2019; Chong et al., 2021; Roy et al., 2021; Worthy et al., 2021), a Poisson distribution, (Koop et al., 1997; Alpert and Knopf, 2016; Kaufmann et al., 2017; Knopf et al., 2020; Yun et al., 2021), or that droplet freezing follows a binomial distribution (McCluskey et al., 2018; Suski et al., 2018; Gong et al., 2019, 2020; Wex et al., 2019). In other cases, authors have used a model of ice nucleation to simulate their experiments and use that simulated distribution to estimate the uncertainty present in their experiment. In the simplest case, droplet freezing is modelled as a Poisson point process (Vali, 2019; Jahl et al., 2021; Fahy et al., 2022b). In more sophisticated models, random variables such as the number of sites, mass of material, and temperature variations are parameterized to run completely new simulated experiments (Wright and Petters, 2013; Harrison et al., 2016). Even in these models, either additional measurements are required, or assumptions must be made about the distribution of each variable. Until the inherent variability behind ice nucleation can be measured to prove or disprove the assumptions being made, all the above methods are only as reliable as the

assumptions themselves. In Section 4, each method, their required assumptions, and the validity of those assumptions are discussed in further detail.

      Empirical bootstrapping is an alternative approach to estimating statistics for a dataset that to our knowledge has not been applied in the context of ice nucleation. In this technique, a series of random samples of the measured dataset is taken to generate estimated statistics that converge on the actual values as the number of samples increases (Efron, 1979; Shalizi,

forthcoming). No assumptions are required about the distributions of random variables underlying ice nucleation and it can be applied to any system where the freezing temperatures or times of droplets are measured. Here we present a set of generalized and statistically rigorous methods based on empirical bootstrapping for quantifying uncertainty in IN spectra. When accompanied by interpolation methods presented in Section 3, this approach can be used to calculate continuous confidence bands and statistically test differences between IN spectra as shown in Section 5. We also address the effects of interpolation

techniques, droplet sample size, and bootstrap sample size to direct the field towards more rigorous and repeatable methods of experimentation and data analysis. An implementation of all presented statistical methods along with documentation and instructions for its use is provided freely for use or reference to assist in future research and improve the statistical treatment of ice nucleation data in the field.

## 2. Sample data and preprocessing

To demonstrate the statistical methods described here, we selected an example IN dataset shown in Fig. 1. The Fuego ground PM37 sample (FUE) from Jahn et al. (2019) was tested for ice nucleation activity before and after being exposed to water in a 1 wt% suspension and allowed to dry under a constant 1 Lpm flow of pre-dried lab air similarly to Fahy et al. (2022b). In both cases, a 0.1 wt% suspension of unaged or aged ash was created in water (HPLC grade, Sigma) filtered through a 0.02 micron pore size Anatop syringe filter. These suspensions were then tested for IN activity on the CMU-CS droplet-on-

substrate system described in detail by Polen et al. (2018) and are compared to a background freezing spectrum obtained from the filtered water used to create the suspensions. Approximately 50 100 nL droplets (1.5 mm diameter) were tested per array with a cooling rate of 1 °C per minute. Two separate suspensions were tested for the unaged ash sample, and three suspensions were tested from ash exposed to water in two separate experiments for the aged ash sample. The previously-determined Braunner-Emmett-Teller (BET) specific surface area of 1.6394 $m^2$ $g^{-1}$ was assumed for all samples.

Since multiple freezing experiments were performed on nominally identical samples (e.g., the replicate suspensions of the same ash or aging experiment), these spectra were combined by merging the lists of freezing events that occurred in each experiment. The frozen fractions and ice nucleation active site density spectra were then recalculated as if the combined freezing events occurred in a single experiment. This is only valid when the IN spectrum of a given suspension is insignificantly different from the combined spectra of all other suspensions and the physical and chemical properties (e.g., suspension

concentration, sample type, water purity, background freezing) are identical between suspensions. The second condition can

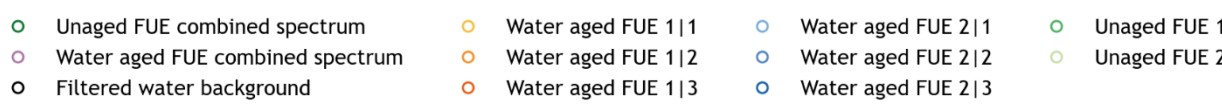

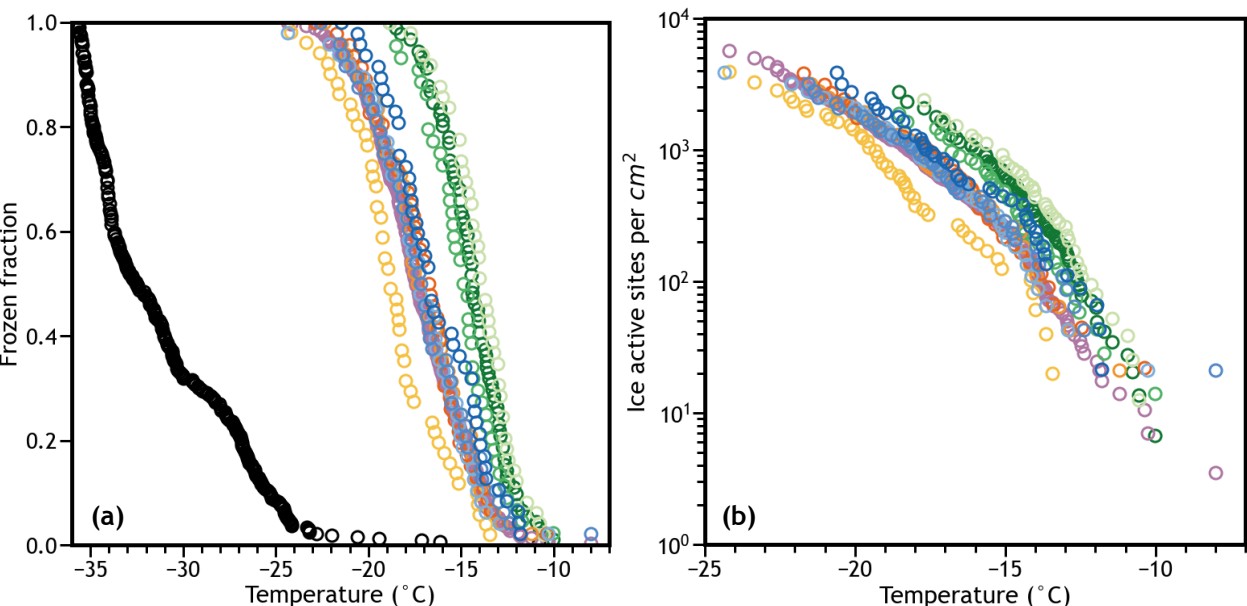

**Figure 1.** Raw (not interpolated or binned) and combined raw a) frozen fraction and b) surface area normalized ice nucleation active site density spectra for sample droplet freezing data used in this study. Water aged spectra are from two separate aging experiments with three freezing experiments each for a total of six individual water aged runs denoted by 'experiment#|freezing run#'

easily be tested in the laboratory, while the first condition can be evaluated using statistical tests described in this paper (see Section 5.2).

The ice active site density spectra were calculated directly based on Eqs. 1-3 (Vali et al., 2015; Vali, 1971, 2019), where $k$ is the differential spectrum, $K$ is the cumulative spectrum, $f(T)$ is the raw fraction of droplets frozen at temperature $T$, $N_0$ is the total number of droplets in the array, $N$ is the number of droplets that have frozen up to a given point, $\Delta N$ is the number of droplets that have frozen in each measurement interval. The variable $\Delta T$ is the size of the measurement interval, the choice of which is discussed below. The normalization factor $X$ can be average droplet volume ($V_d$), mass of sample suspended per droplet ($m_d$), or as is used here, specific surface area of sample suspended per droplet (Usually BET specific surface area; $BET_{SSA}$), giving the number of ice nucleation sites that are active at temperature $T$ per unit of suspension volume (usually denoted $K$), sample mass (denoted $n_m$), or sample surface area (denoted $n_s$) respectively. The derivation of these equations requires that $X$ be identical for every droplet being analyzed – an important assumption and source of error. However, as will be discussed later, the empirical bootstrapping approach quantifies this source of error, meaning these parameters can be used and interpreted even when the assumption does not strictly apply if the uncertainty is also incorporated into the interpretation.

$$k(T) = -\frac{1}{X\Delta T} ln\left(1 - \frac{\Delta N(T)}{N(T)}\right) \tag{1}$$

$$K(T) = \frac{1}{X} \ln \frac{N_0}{N(T)} = -\frac{1}{X} \ln(1 - f(T)) = \sum_{T_i}^{T_f} k(T) \, \Delta T \qquad (2)$$

$$n_s(T) = -\frac{1}{BET_{SSA} m_d V_d} \ln(1 - f(T)) \qquad (3)$$

Differential IN spectra have only recently come into common use because most interpretations of the formulation of *k* require high-quality data (e.g., hundreds of uniformly sized droplets with low background freezing activity in our estimation) for the coarse binning process used to ensure the data are not too sparse. See Vali (2019) for an in-depth discussion of this approach based on improvements in ice nucleation data quality obtained using droplet microfluidics by Polen et al. (2018) and Brubaker et al. (2020) that make the application of differential IN spectra feasible. However, differential spectra are extremely useful both for visual interpretation of data and for quantitative comparison of IN spectra. Specifically, they can provide information on how many IN sites become active at a given temperature, which is not immediately obvious from examining a cumulative spectrum. For a more generally useful method of directly calculating raw differential spectra, $\Delta T$ was chosen such that the endpoints of the temperature interval for a datapoint were the midpoints of the distance between the nearest neighbors on either side of the datapoint, and $\Delta N$ is the number of droplets that froze at that datapoint. Vali (2019) stated that this approach loses the quantitative significance of *k* because the value of *k* will vary based on the size of $\Delta T$, however, we contend that it is this variation in the size of $\Delta T$ that maintains the quantitative aspect of *k*, as the magnitude will be inversely proportional to the number density of freezing events with respect to temperature as expected. This results in noisy data, but when coupled with the interpolation techniques presented in Section 3, this problem can be resolved using a smoothing algorithm. This approach has the advantage of ensuring that every 'bin' has at least one freezing event in it while maintaining the advantages of differential IN spectra compared to cumulative spectra, even for relatively low-quality data. However, as will be shown in Sections 4 and 5, high-resolution data (e.g., from hundreds of droplets and/or several freezing experiments) are still required for statistical comparisons of differential IN spectra.

## 3. Interpolating freezing curves and calculation of continuous spectra

The most common style of reporting ice nucleation activity is using the cumulative ice nucleation active site density curves calculated directly from raw data as shown in the previous section, but there is an important limitation to this type of data treatment. While it represents the data exactly as measured, there is no way to quantitatively compare one raw freezing spectrum with another without some type of interpolation. This is because even if a droplet freezes at a particular temperature in one experiment, there is no guarantee that a droplet will freeze at or near that temperature in another experiment. Often the approximate difference between spectra is just compared by eye for lack of a better method. This presents issues when trying to subtract a background spectrum or when quantifying uncertainty and testing statistical difference between spectra and leads to a need for effective interpolation methods for comparing IN spectra.

### 3.1 Binning and its shortcomings

One common method for interpolating IN spectra is through temperature binning, where a temperature interval is represented by a single value of IN activity that is treated as constant throughout the interval. This approach is appealing, as it aligns with the discrete nature of IN experiments and allows straightforward calculation of differential IN spectra by using the bin width as $\Delta T$ (Vali, 2019). However, binning is widely accepted as an inefficient interpolation method for measurements of continuous variables such as ice nucleation activity and has been shown to reduce statistical power and bias statistical results

in data from a variety of disciplines (Gehlke and Biehl, 1934; Selvin, 1987; MacCallum et al., 2002; Altman and Royston, 2006; Manley, 2014; Virkar and Clauset, 2014; van Leeuwen et al., 2019). While ice nucleation activity is measured discretely, it is a continuous property – any given ice nucleation site has a theoretical ice nucleation rate over the entire continuous temperature range and combining many ice nucleation sites together results in a continuous curve, sometimes with multiple critical or inflection points (Beydoun et al., 2016). It is therefore desirable to transform the discrete measurements into

continuous space to accurately represent IN activity rather than further discretizing them as in binning.

### 3.2 Previous methods of continuous functional interpretation

     To make a discrete variable continuous, some type of functional interpolation is required. Many studies approximate IN spectra as exponential polynomials or similar simple functions (Atkinson et al., 2013; Kanji et al., 2013; Niedermeier et al., 2015; Harrison et al., 2016, 2019; Peckhaus et al., 2016; Vergara-Temprado et al., 2017; Price et al., 2018). Exponential

polynomials can capture the overall exponential shape of cumulative IN spectra in most cases, however, they impose explicit assumptions about the shape of the IN spectra through their closed-form expressions. Particularly in samples that contain mixtures of different types of ice nucleation sites (e.g., Beydoun et al., 2017), simple polynomials are likely to be insufficient for accurate interpolation of IN spectra.

     Instead, the ideal interpolation method would take a series of measured datapoints from a droplet freezing experiment

and would output a continuous IN parameterization that could predict the IN activity of the sample at any temperature. A parameterization such as a contact angle scheme (Chen et al., 2008; Beydoun et al., 2016; Ickes et al., 2017) or the singular-stochastic formulation of ice nucleation (Vali, 2014; Barahona, 2012; Niedermeier et al., 2011) would be preferred, however, these parameterizations require preexisting knowledge or assumptions about of the nature of the sample being tested. For data analysis in laboratory or field studies, this information is often not available, and we must look for an interpolation method

that can capture an ice nucleation spectrum with any shape.

### 3.3 Piecewise interpolation for ice nucleation spectra

     For a generally applicable interpolation scheme, piecewise fitting algorithms such as spline fit all requirements. Spline fits provide interpolations of arbitrarily complex data by fitting a series of polynomials to small portions of the available data. The resulting piecewise functions are continuous and differentiable, meaning that only one or the other of the cumulative or

differential IN spectrum must be directly fit from the data – the other spectrum can be calculated by either computing the negative derivative of the cumulative freezing curve or the negative antiderivative of the differential spectrum. To find the fitting method that performs well, a variety of algorithms available in the Python Scipy library (Virtanen et al., 2020) were modified and tested for their ability to interpolate the combined water aged volcanic ash ice nucleation spectrum. 'Splinederiv' uses a cubic spline fit of the cumulative spectrum, 'splineint' uses a cubic spline fit of the differential spectrum, 'PCHIP' uses the piecewise cubic Hermite interpolated polynomial algorithm of the cumulative spectrum, and 'smoothedPCHIP' is the PCHIP curve followed by a cubic spline fit with a smoothing factor.

Figure 2a (cumulative $n_s$) and 2b (differential $n_s$) compare these methods to a binning approach and the raw data from Fig. 1 using the water aged FUE ash spectrum. Note that the interpolated spectra do not start until there is a sufficient density of freezing events (more than one per degree Celsius) to avoid overfitting and because the error on these initial points is much larger than that of the rest of the spectrum as will be seen later. On initial inspection, basic spline fits perform well at higher ice active site densities. However, the splinederiv algorithm does not always maintain the monotonicity requirement intrinsic to the cumulative spectrum (and correspondingly are not strictly positive in the differential spectrum). The splineint algorithm corrects for this, but performs relatively poorly in capturing the behavior of early freezers, overestimating the IN activity between –10 and –14 °C. The solution to these two problems is to interpolate the cumulative spectrum with the monotonicity constraint offered by the PCHIP fitting algorithm and to take the derivative of this interpolation for the differential spectrum. This method reproduces the shape of the cumulative IN spectrum because it calculates an exact interpolation, but as a result is extremely noisy in the differential spectrum. By adding an additional smoothing step after the PCHIP interpolation (using a simple smoothed cubic spline fit after the PCHIP algorithm), a smooth and interpretable interpolated ice nucleation spectrum can be derived from the raw data without losing the detail present at the high/warm temperature end of the spectrum visible in the differential plot as shown in the smoothedPCHIP spectrum. The smoothedPCHIP curve is monotonic and accurate to the observed data in the cumulative spectrum and is smooth and readable in the differential spectrum and will be used for the remainder of this work. In Fig. S1 of the Supplemental Information (SI), the smoothedPCHIP algorithm is applied to each individual volcanic ash IN spectrum, and in Fig. S2, it is applied to the combined unaged and water aged spectra to compare the interpolations with their corresponding raw datapoints.

## 4 Calculating confidence intervals and bands

### 4.1 Elementary statistical methods

The question of how to calculate confidence intervals for IN spectra derived from droplet freezing experiments has been addressed several times in the IN literature. In some cases, a normal distribution about the frozen fraction curves is assumed. Where multiple freezing experiments are available and are interpolated such that means and standard deviations can be calculated for a collection of freezing spectra, a Z-interval (based on the normal distribution) or t-interval (based on Student's t-distribution) can be constructed (Polen et al., 2018; Jahn et al., 2019; Worthy et al., 2021) or standard deviations

and standard errors are sometimes reported as-is (Losey et al., 2018; Chong et al., 2021; Roy et al., 2021). While it is unclear how many droplets and replicate freezing assays are required for these approximations to be valid under the Central Limit Theorem, it is unlikely that most existing freezing assay datasets achieve this sample size requirement, since confidence intervals calculated using these techniques often disagree with those calculated using other methods described below and those presented in this study. It is also unclear what exactly a required sample size would mean in this context: the number of droplets is not sufficient, because each droplet does not contribute to every point on the observed ice nucleation spectrum equally. However, the number of separate ice nucleation assays is also not sufficient, as techniques that measure hundreds of droplets in a single assay should require fewer overall assays to calculate accurate statistics because there are more droplets contributing to the accuracy of each point on the measured ice nucleation spectrum. Some combination of the two is required, but there is no existing method by which the accuracy of confidence intervals for an ice nucleation spectrum can be evaluated based on the relevant sample sizes."

Other studies (e.g., McCluskey et al., 2018; Suski et al., 2018; Wex et al., 2019; Gong et al., 2019, 2020) have calculated approximate confidence intervals for frozen fraction values by treating them as binomial ratios and using the adjusted Wald interval suggested by Agresti and Coull (1998). In the latter case, calculating uncertainty for derived ice active site density spectra requires propagation of error through Equations 1 and 2, followed by an assumption of normality when the confidence intervals are calculated. There is, however, no reason to believe that the spread of freezing events in droplets should even approach a normal distribution, making this assumption unreliable,

A better approximation for the variability in droplet freezing experiments is the Poisson distribution, in part because the widely used ice active site density spectra are based in Poisson statistics (Vali, 1971), but also because droplet freezing resembles a Poisson point process where freezing events occur approximately continuously and independently at a given rate. Koop et al., (1997) suggested the use of Poisson fiducial limits to calculate uncertainty in a variety of types of freezing experiments, and this approach has been used by several studies since (Alpert and Knopf, 2016; Kaufmann et al., 2017; Knopf et al., 2020; Yun et al., 2021). However, the distributions of IN sites across particles, distributions of these particles among droplets, distributions of freezing abilities of individual IN sites, distributions of freezing events that occur based on the aggregate freezing ability in a droplet, and temperature distribution between the droplets could all serve to skew or otherwise change the distribution of droplet freezing events measured. Using a Poisson distribution corrects for only some of these random factors, and because ice active site spectra are based on the Poisson process, these are the variables that most need to be considered when calculating experimental uncertainty. Thus, while these closed-form confidence limits are convenient, they are not likely to be accurate.

## 4.2 Parametric bootstrapping and its shortcomings

Another class of methods of calculating confidence intervals for freezing spectra relies on a technique known as bootstrapping, where artificial freezing experiments are generated from a measurement using Monte Carlo simulations (Davison and Hinkley, 1997). When the simulations are based on an existing ice nucleation theory (e.g., when simulated

experiments are produced using a parameterization of ice nucleation), this technique is known as parametric bootstrapping, and given enough simulations, the artificial experiments represent the full range of possible variability around the measured result that could be observed in the theoretical framework used.

        For example, based on Wright and Petters (2013), Harrison et al. (2016) and subsequent publications simulate a number distribution of ice active sites in a collection of theoretical droplets based on the ice active site densities calculated

from the original experiment. This model can be used to simulate freezing spectra by sampling these theoretical droplets and assuming that freezing events occur when the number of ice active sites in each droplet is greater than or equal to one. When repeated enough times, this distribution of freezing spectra can be used to calculate confidence intervals for the measured data either by assuming that the quantiles of the distribution of simulated freezing spectra approximate the confidence intervals or by calculating simple Z-intervals from the distribution of simulated freezing spectra (although the latter invokes an assumption

of normality).

        An alternative method of parametric bootstrapping for confidence intervals of IN spectra models individual droplets freezing as a Poisson point process (again the same assumption used in deriving ice active site density spectra) as shown in Vali (2019) and applied in Jahl et al. (2021) and Fahy et al. (2022b). In this approach, the number of droplets that freeze in each temperature interval (or equivalently, the rate of droplet freezing) is used as the mean value of a discrete Poisson

distribution. Then, for each temperature interval, a new number of droplets freezing in the interval is selected from the distribution. When this is done for all temperature intervals, the simulated values are combined into a simulated experiment. Once ice active site density spectra are calculated from these simulations, and this process is repeated 100s to 1000s of times, the quantiles of the distribution of simulated ice active site densities for each temperature bin can be used as an approximation of confidence intervals.

Both parametric bootstrapping approaches described here rely on the parameterization to produce accurate results, meaning that if the parameterizations are approximate or inaccurate, they may produce misleading or incorrect statistics. An in-depth analysis of the accuracy of the assumptions of each of these parameterizations is beyond the scope of this paper, but there are major concerns for each model. The calculations based on particle distributions in droplets (Wright and Petters, 2013; Harrison et al., 2016) assume that ice active sites are distributed evenly across the surface of a material, that the material is

suspended evenly throughout the droplet, and possibly (depending on the specific approach) that the material is composed of uniform spheres and that ice nucleation is time-independent or the characteristic temperatures for each given ice nucleation site are normally distributed. The first assumption is known to be false for some materials; minerals often have higher concentrations of and/or more ice active IN sites near or in specific nanoscale defects, cracks, pores, or other specific regions such as the perthitic textures in some feldspar minerals (Whale et al., 2017; Kiselev et al., 2017; Holden et al., 2019; Friddle

and Thürmer, 2020). The second assumption may or may not be true, especially at higher suspension concentrations (Beydoun et al., 2016). The third assumption depends on the material in question. The fourth assumption ignores time, one of the most important factors introducing uncertainty and randomness into droplet freezing experiments (Wright and Petters, 2013; Herbert et al., 2014; Vali, 2014; Knopf et al., 2020), and the fifth assumption does not have a theoretical basis and requires additional

experimentation to determine the parameters of the normal distribution (Wright and Petters, 2013). Regardless of the specific approach used, these techniques either require extensive experimentation to determine the nature of the ice nucleation material being studied or rely on assumptions that produce an incomplete and potentially inaccurate parameterization.

The calculations based on the Poisson distribution (Vali, 2019; Fahy et al., 2022b; Jahl et al., 2021) have very different assumptions. Stochasticity and IN site variability are accounted for in the process of simulation from the measured IN spectrum, however, this method requires coarse binning, as ideally multiple freezing events will occur within each bin. As discussed before, binning continuous data is inefficient. It also assumes that in these bins, the nucleation rate does not change with temperature. For coarse temperature bins especially, this assumption will break down, as ice nucleation spectra are strong exponential functions of temperature (Fletcher, 1969). While the Poisson parametric bootstrapping method makes fewer assumptions and captures more variability than other parametric methods, it relies on risky and/or false assumptions, contributing systematic error to the confidence intervals. Note it is not the purpose of this study to quantitatively compare methods previously used to calculate uncertainty in IN spectra, and the above discussion is only as a qualitative overview of the assumptions and approximations previous methods use.

## 4.3 Empirical bootstrapping and its benefits

The other class of bootstrapping method, non-parametric bootstrapping (known as empirical bootstrapping), does not rely on any parameterizations. Instead, the original experimental data and sampled from with replacement (i.e., the same datapoint can be sampled more than once) to produce artificial datasets (Efron, 1979; Efron and Tibshirani, 1994; Davison and Hinkley, 1997; Shalizi, forthcoming). This method is remarkably well-suited to the problem of ice nucleation statistics, as droplet freezing experiments result in a list of freezing temperatures that can be easily sampled from to create new simulated droplet freezing experiments. The large droplet numbers coupled with a limited freezing temperature range ensure that the empirical data covers most of the possible variability within each experiment. If multiple freezing experiments are performed on identically prepared samples, this method will even capture the variability in sample preparation and other aspects of the experiments being performed. Since variations in droplet size, sample mass suspended, or distributions of surface area among droplets (the parameters behind the normalization constant $X$) also contribute to the variability observed in experiments, the error caused by assuming $X$ is constant between droplets is also included into the model. Empirical bootstrapping requires no physical model of ice nucleation and so captures the stochastic nature of ice nucleation, inhomogeneities in ice active site distributions within the sample, and even any potential unknown sources of variability within IN active materials. Thus, empirical bootstrapping is universally applicable, can capture all sources of variability in an experiment, and is unambiguous in its implementation, making it an ideal candidate for a standard statistical method for analyzing ice nucleation experiments. The only assumption required (which has already been made when deriving ice nucleation spectra) is that all datapoints must be statistically independent, meaning that no droplet can affect any other droplet's freezing temperature (Shalizi, forthcoming). This condition is already required for accurate ice nucleation measurements and is already implemented in most laboratories by physically isolating droplets using an inert oil or gas or by separation of droplets in wells or microwells. Empirical

bootstrapping is only otherwise limited by the computational time available to draw many statistical simulations of an ice nucleation spectrum and the quality of the observed data itself (Hesterberg, 2015), both of which are addressed further in Section 4.6.

05    Figure 3a and 3b show the application of empirical bootstrapping to simulate cumulative and differential spectra for the combined and interpolated volcanic ash ice nucleation data previously introduced in Figs. 1 and 2. Each spectrum is statistically simulated by randomly sampling with replacement $n$ times from the list of freezing temperatures using a discrete uniform distribution function in the original experiment using the choices function in the built-in random library in Python, where $n$ is the number of droplets in the original experiment. Where multiple droplets froze at a given temperature, that

10 temperature is added multiple times to the 'observed' list. This process is repeated to create new 'sampled' freezing temperature lists until the desired number of simulated experiments is achieved. Each sampled list is then sorted and analyzed as distinct freezing assays, each with its own IN spectra that can be interpolated as usual. The simulated spectra are distributed around the true data due to variations in which droplets are randomly sampled for each simulation, and the width of this distribution provides an estimate of how uncertain the experimental value is at that temperature. Summary statistics of this distribution

15 such as mean, standard deviation, and quantiles can be calculated by dividing the temperature range of each interpolated spectrum into a dense grid of evenly spaced points (e.g., ~10 points per degree Celsius) and calculating each statistic as usual using the distribution at each point. The resulting statistics as a function of temperature can then be interpolated exactly using a simple spline fit due to the high density of data available, providing interpolated continuous functions for each summary statistic. This process is detailed further in the SI.

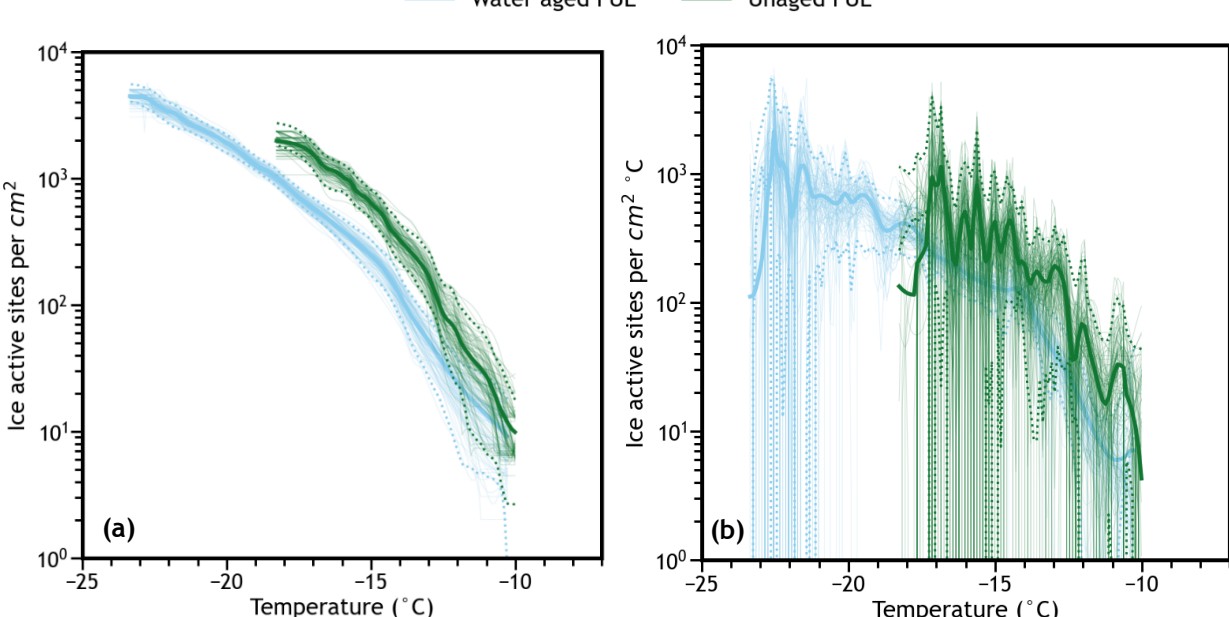

**Figure 3.** Interpolated combined data (bold line), interpolated 2.5th and 97.5th quantiles (dashed lines), and interpolated individual simulations (faint lines; N=100) of the a) cumulative $n_s$ and b) differential $n_s$ spectrum for each set of combined ash data from Fig. 1. The simulated data (lines) using empirical bootstrapping provide a realistic estimate of the distribution of how the spectra could vary based on stochasticity, variations in the individual droplet freezing experiments comprising the combined experimental spectrum, and other factors contributing to experimental uncertainty.

While the mathematical theory behind empirical bootstrapping is complex (see Efron and Tibshirani, (1994) or Davison and Hinkley, (1997) for a thorough treatment of the mathematics behind bootstrapping and Canty et al., (2006) for a thorough discussion of inconsistencies and errors that can be encountered when using bootstrapping), Fig. 3 provides some evidence that this approach has successfully captured the possible variability in the ice nucleation spectra. Using the interpolated quantiles as a measurement of the spread of the simulated spectra, the magnitude of the variability in each spectrum

largely follows the trends that would be expected. For example, the simulated cumulative spectra have much less relative variability than the simulated differential spectra and both types are less variable at intermediate temperatures where more droplets froze in the actual experiments. This reflects the fact that increased sample sizes tend to reduce uncertainty as cumulative spectra represent a sum of all previous datapoints, and most droplets tend to freeze at intermediate temperatures in a droplet freezing assay. The noisiness of the differential spectra indicates large uncertainty, meaning the differential spectrum

for the unaged volcanic ash is largely uninterpretable, while the differential spectrum for the aged volcanic ash and both cumulative spectra are much more descriptive – for example, it can clearly be seen that the two cumulative spectra do not overlap significantly below –13 °C, a fact that will be further quantified in Section 5.  The large variability (even to $k$ values of zero in some cases) observed at the extrema of both differential spectra represent relatively rare freezing events for the samples, such that a given simulation may or may not observe freezing events at that temperature. These areas are more

common in the unaged volcanic ash IN spectrum because that spectrum consists of fewer droplets than the aged volcanic ash

IN spectrum, highlighting the importance of high-resolution data for meaningful interpretations of differential spectra. Also note that in the last degree of each differential spectrum, the measured results lie outside of the quantiles calculated because 100 simulations are not sufficient to fully estimate the variability in the measured spectrum; this problem would be remedied with additional simulations that fully sample that region of the IN spectrum.

**4.4 Basic bootstrapped confidence bands and their limitations**

Using this new method to simulate data that capture the variability inherent to freezing experiments, bootstrapped summary statistics describing the experimental measurement can be calculated. Values such as the bootstrapped standard error of the mean approximate the true standard error of the mean remarkably accurately when large numbers ($n \geq 200$) of bootstrap simulations are employed, a fact known as the 'plug-in-principle' (Efron and Tibshirani, 1994). Bootstrapped confidence
intervals, however, are a more difficult subject. The previous studies that used parametric bootstrapping methods assumed that the $\alpha/2$th and $1-\alpha/2$th quantiles of the simulations correspond to the lower and upper limits of the $1-\alpha$ level confidence interval respectively, where $\alpha$ is the threshold value chosen for statistical significance (Harrison et al., 2016; Vali, 2019). This assumption is common, and while it can work well for many applications, this 'quantile interval' has little basis in statistical theory. It is also strongly affected by bias, only partially corrects for skewed distributions (ice nucleation spectra are likely to
be skewed upward based on the Poisson statistics they are derived from) and can be too narrow when applied to some distributions (Hesterberg, 2015; Efron, 1987). The strong bias that quantile intervals exhibit is particularly concerning when using potentially inaccurate parametric bootstrapping or when a small sample results in poor sample coverage in empirical bootstrapping.

Fortunately, other bootstrap confidence intervals exist. For a simple interval rooted in statistical theory, we can
construct the reverse percentile interval, also known as the pivotal interval, where the upper and lower quantiles are subtracted from twice the sample mean for the lower and upper confidence intervals respectively. However, in skewed distributions such as uncertainty in ice nucleation spectra, the pivotal interval tends to be inaccurate. For a more traditional interval, we can construct a Z-interval around the measured spectrum with a bootstrapped estimation of standard error, but this assumes a normal variance – obviously a poor approximation of the complexity inherent to ice nucleation. A bootstrapped t-interval
(tboot) using the number of droplets in the original experiment as the number of degrees of freedom is a slightly better estimate, but still suffers from error from bias (including narrowness bias) and skewness (Hesterberg, 2015; Efron, 1987).

## 4.5 Better bootstrapped confidence bands

Significant work has gone into correcting these problems with basic bootstrapped confidence intervals. The tboot interval can be corrected for skewness to the 'tskew' interval by including a second-order skewness term in the tboot calculation as shown by Johnson (1978). The quantile interval can be expanded by changing the quantile bounds by a factor related to the t-statistic to remove narrowness bias, called the 'expanded quantile interval' or the BCa confidence interval (Efron, 1987; Hesterberg, 2015). However, by far the most accurate method is the studentized confidence interval, referred to as the 'bootstrap T' or 'confidence intervals based on bootstrap tables' elsewhere (Efron and Tibshirani, 1994; Hesterberg, 2015; Diciccio and Efron, 1996; Efron, 1979). This method corrects the errors of the t-interval by estimating the actual distribution of the t statistic for through bootstrapping. Specifically, the standard error of each simulated spectrum is calculated and is used to normalize the difference of each simulated spectrum from the mean of all simulated spectra. These normalized values are compiled into another distribution and the desired quantiles of this distribution are multiplied by the standard error of the collective of simulated spectra to produce the final confidence intervals. To obtain the standard error of each individual simulated spectrum a second round of bootstrapping is needed using the simulated spectrum as the 'true' measurement resulting in 'resimulated' spectra. Further descriptions and equations for calculating this and all previously mentioned confidence intervals are provided in the SI. The process is computationally intensive, but it results in confidence intervals that are

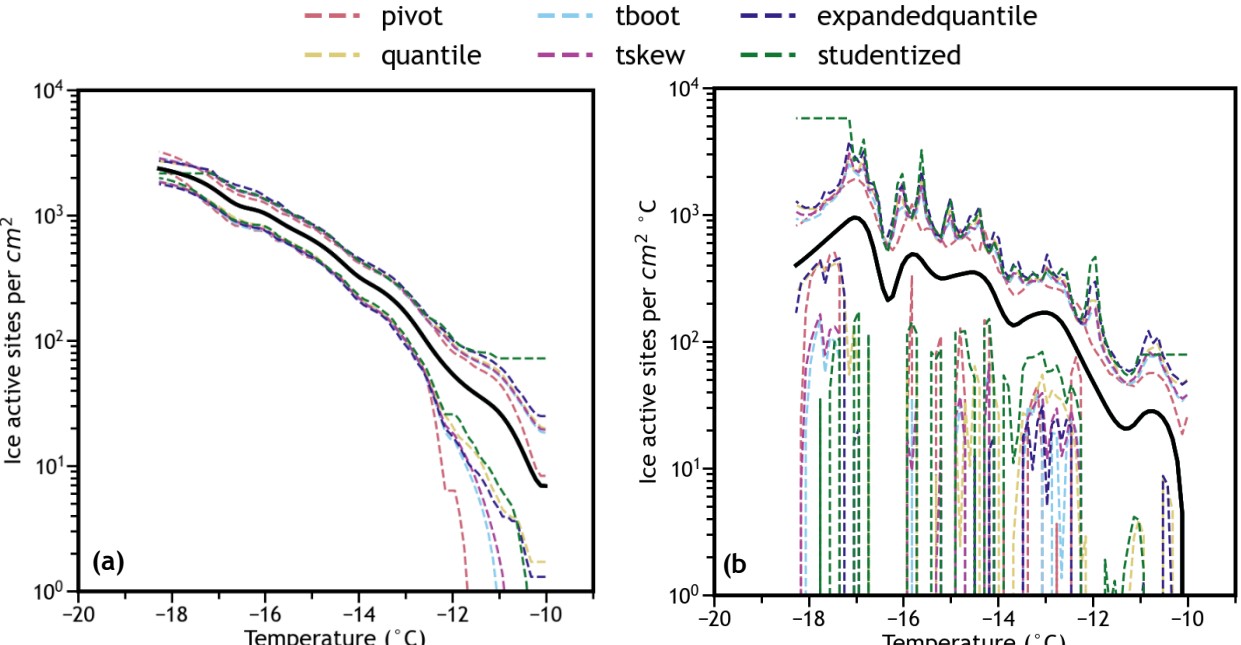

**Figure 4.** Comparison of methods to calculate confidence bands (shown as different-colored dashed lines) for a) cumulative and b) differential unaged volcanic ash $n_s$ spectra representing 91 droplets. Experimental spectra are shown in black. Most calculation techniques are in good agreement across the range of the cumulative spectrum, but all techniques except the studentized intervals (green dashed line) overestimate the variability in the differential spectra, resulting in confidence bands that are too wide across most of the spectrum.

unaffected by bias or skewness, even for small sample sizes. Statistical theory calls such intervals second-order accurate, and this property sets the bootstrap T apart as a standard to compare other confidence intervals against.

The above methods were used to calculate confidence bands (continuous confidence intervals) for the cumulative and differential IN spectra of the unaged and water aged combined volcanic ash sample. Like the summary statistics, to create confidence bands confidence intervals were calculated at every 0.1 °C within each spectrum and were interpolated using a smoothed cubic spline fit. The different methods of calculating confidence bands for the combined unaged spectra representing 91 droplets (the number of droplets present in the two unaged FUE freezing experiments) are compared in Fig. 4. In general, the best-performing confidence bands (determined by being the closest to the studentized bands) are calculated using the

quantile or expanded quantile methods and the skew-corrected t-interval method – the other approaches tend to be less accurate. In contrast, in the combined water aged spectra representing 286 droplets, the different methods of calculating confidence bands are in good agreement over most of the IN spectra (Fig. S3), although the studentized confidence bands show slightly different behavior at high temperatures where very few droplets are observed to freeze. As in Fig. 3, the variability in the differential spectrum for this relatively low-resolution data is significant as shown by the wide confidence bands in Fig. 4b,

although most confidence bands overestimate this variability compared to the studentized bands.

**4.6 Recommendations for use of empirical bootstrapping methods and required sample sizes**

If accurate confidence bands on both the cumulative and differential spectra are required from low-resolution data, studentized intervals should always be used. Ideally, the studentized confidence bands should be used in all cases, but the computational time required for calculation of these confidence bands can be excessive. For most use cases then, the tskew

bands are somewhat conservative confidence bands rooted in theory, and we will use them in the remaining examples below. Expanded quantile or quantile bands are also an appropriate choice when empirical bootstrapping is used but should be tested against the studentized bands for each system to check for potential biases in the data collection process. Quantile bands should be avoided when using small numbers of droplets (>5 per degree Celsius measured seems to be acceptable) or when using a parametric bootstrap, as the biases inherent to the parameterization will be amplified by the quantile bands. The pivot and tboot

00 bands seem to be poor choices in the context of ice nucleation.

Although we cannot theoretically determine the sample sizes required for accurate confidence bands using empirical bootstrapping due to the same limitations discussed previously, the sample sizes required for accurate confidence bands can be empirically evaluated by testing how many assays, droplets, and simulated spectra are required for confidence bands to converge (therefore reducing the uncertainty of the confidence bands due to sample size). Fig. 5a displays interpolations and

05 resulting confidence bands for the differential IN spectrum of aged volcanic ash when 50, 100, 150, 200, and 286 (where all droplets are included) droplets are randomly sampled from the six performed experiments. The width and shape of the confidence bands changes significantly but seem to be converging to a smooth curve exemplified when N=286. When N=50, the confidence bands span three or more orders of magnitude, indicating that 50 droplets may be too few to draw any conclusions from freezing spectra in our system. In this case, the minimum sample size is approximately 150 for useful

conclusions at the 95% confidence level, and at least 200 is preferred (at least in our system) for more accurate confidence bands to ensure the entire probability space of droplet freezing is covered. More droplets or freezing assays improve the accuracy and reduce the width of the confidence bands, especially in differential IN spectra, further motivating the many recently developed microfluidic techniques (Stan et al., 2009; Weng et al., 2016; Reicher et al., 2018; Tarn et al., 2018; Brubaker et al., 2020; Roy et al., 2021). Additionally, the number of simulations (and resimulations if using studentized

 confidence bands) should be chosen carefully to ensure the full variability present in IN spectra is represented.

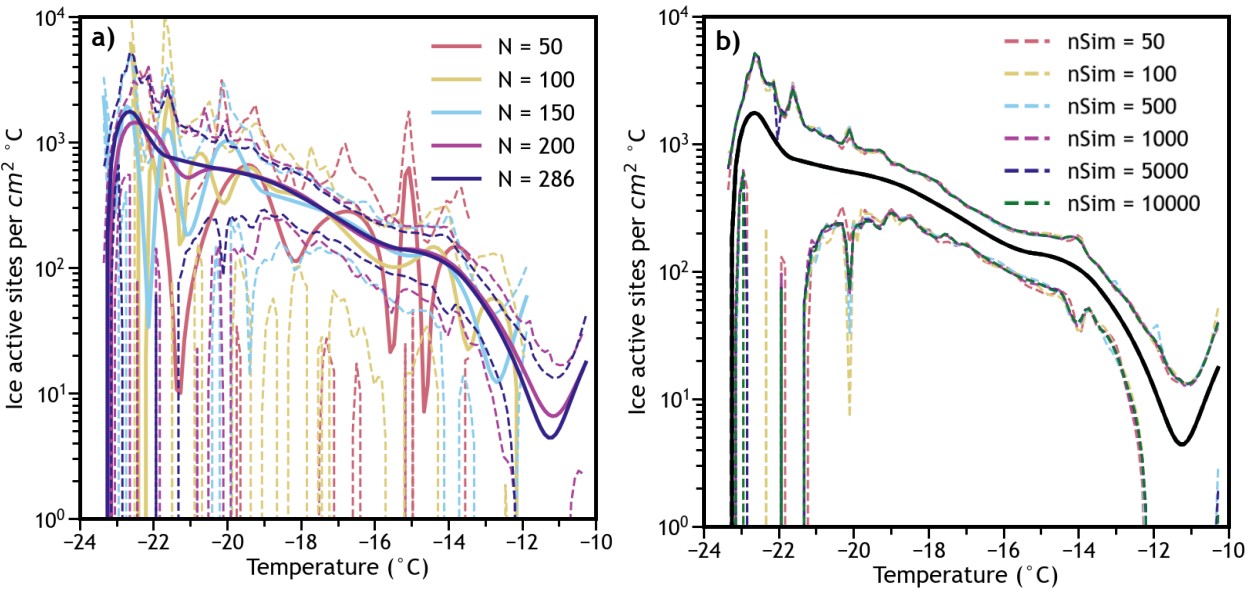

**Figure 5.** Differential freezing spectra of the water aged FUE ash with tskew confidence bands a) calculated with varying numbers of datapoints randomly sampled without replacement from all six experiments with 1000 bootstrapping simulations, and b) calculated using different numbers of bootstrapping simulations, with the experimental combined spectrum shown in black. Figure S4 shows the cumulative spectra, but the effects of sample size are not as pronounced.

In Fig. 5b, the tskew confidence bands of the combined water aged volcanic ash IN spectra (all 286 droplets) are compared when the number of simulations (*nSim*) ranges from 50 to 10000. S4 shows the same bootstrap simulation number analysis when using 50 and 150 droplets randomly sampled from the initial 286 to test the effects of droplet number on the required bootstrap simulation sample size. Based on these plots, the number of bootstrapped spectra does not appear to impact

 the confidence intervals nearly as much as the number of droplets used. This is a contrast from requirements in other types of bootstrapping techniques, which can be easily calculated to require more than 10000 samples to be accurate within an acceptable margin of error of 10% (Hesterberg, 2015). While this insensitivity could be coincidence (errors in opposite directions cancelling out to result in confidence bands that are approximately correct), we speculate it is instead related to the fact that each bootstrap sample includes many droplets (286 in this case) which are also sampled, covering the probability

 space more completely than when single datapoints are sampled and therefore reducing the Monte Carlo error observed. Based on this, we recommend ensuring that the number of simulations multiplied by the number of droplets (the 'resample size') in each simulation exceeds 10000 – in the above spectrum this resample size ranges from around 14000 to 2.86 million. This is

partially corroborated by Fig. S5, where the resample size reaches as low as 2500 and the confidence bands with low resample sizes often do not match those with high resample sizes. It is also important to consider the statistic being calculated – in the above case the standard deviation and skewness are being used to calculate confidence bands, but if quantiles were being used, the number of bootstrapped spectra would have to be large enough to calculate accurate $2.5^{th}$ and $97.5^{th}$ quantiles (likely about 200 spectra). Regardless, the effects of bootstrap sample size should be tested whenever empirical bootstrapping is applied to ensure that the accuracy of the calculated confidence bands (or any other statistic) is never dependent on the number of simulations used. Similarly, each investigator must determine their own droplet sample size requirement to collect datasets that can answer their research questions.

Finally, Figs. 3-5 provide evidence that the interpolation technique used is not overfitting the data, as the quantiles and other confidence bands follow the general shape of the experimental spectra. Since these statistics are calculated from an aggregate of 1000 samples in most cases, they would be expected to smooth out random variation present in a single measured spectrum that could be causing the complex interpolated curve observed. Because the aggregated data maintains the same shape, it can be assumed that it is at least somewhat meaningful, and that the interpolation technique is using an appropriate smoothing factor, however, this should be tested regularly to minimize potential overfitting. Note that when droplet numbers are below 200 (as in some of the Fig. S4a spectra and in Fig. S5) the interpolated differential spectra have shapes that look unrealistic (e.g. many inflection points within one or two degrees Celsius), but they also have error bars that span many orders of magnitude in those regions, meaning that neither the measured value nor the interpolation of the differential IN spectrum at that point is as useful because the uncertainty is so high. Regardless, the interpolation of the cumulative spectrum remains smooth and interpretable, and the portions of the interpolated differential spectrum with lower uncertainties are still meaningful.

## 5 Comparing IN spectra, testing statistical significance, and background subtraction

Confidence bands provide useful information about the variability of a single dataset – in the case of droplet freezing assays, 95% confidence intervals contain the true population mean ice nucleation activity of the suspension being sampled from in 19 out of 20 analyses (that is, either the true spectrum is within the confidence interval, or an event of probability at most 5% happened during data collection). All ice nucleation data should be reported with some form of confidence interval or quantification of the distribution of the measurements (e.g., standard error bars). These statistics must be calculated using a method, such as empirical bootstrapping, rooted in statistical theory to minimize assumptions about the ice nucleation experiment and accurately represent the uncertainty inherent to the experiment.

Another key application of statistics that quantify the variability within a dataset is in comparing measurements of different samples to assess the degree of similarity of their INA. In general terms, confidence bands can be used to compare two IN spectra by determining whether they could reasonably have been drawn from the same population. Often confidence intervals or bands are interpreted based on whether they overlap: If confidence intervals of two spectra do not overlap, they

are statistically significantly different. However, it is not necessarily true that if the confidence bands overlap the two measurements are statistically the same at a given confidence level. This common misconception is based on the difference between error bars calculated using the standard error of the mean and confidence intervals (Barde and Barde, 2012; Belia et al., 2005).

For a more quantitative (and interpretable) method to compare IN spectra can simply be divided or subtracted. We
will use the term 'difference spectrum' to refer to this ratio or difference as a function of temperature, as both are calculated using the same procedures and provide similar information. When interpolated IN activity spectra are used, a continuous difference spectrum can easily be generated by calculating the ratio (or difference) between two interpolations at each point in a dense grid of temperatures, then interpolating between those points. A difference spectrum can be plotted as a function of temperature with its own confidence bands and can be used to test whether two IN spectra are statistically significantly different
at any temperature where the two spectra overlap at any confidence level. Stated precisely, the hypothesis that the two IN spectra are different can be tested against the null hypothesis that the two IN spectra are not quantitatively different. in the case of a ratio-based difference plot with confidence bands, if the confidence bands do not contain one at a given temperature, then the null hypothesis is rejected. If they do contain one, then that claim cannot be made. If a difference between IN spectra is used instead of a ratio, then zero is used for this hypothesis test instead of one. Therefore, if confidence bands can be accurately
calculated for a difference spectrum, then continuous statistically rigorous claims about differences between IN spectra can be tested.

## 5.1 Calculating confidence bands for difference spectra

Calculating confidence bands for differences or ratios of continuous variables is not trivial, but for these metrics to be useful, confidence bands are necessary. Subtracting or dividing the confidence bands of the compared spectra is not accurate.
Elementary propagation of error formulas assumes that the variability within both spectra (and of the comparison spectra) is normally distributed, which is a poor assumption as discussed previously. Again, bootstrapping offers a solution. To simulate the variability in the difference spectra, individual simulations of each measurement can be subtracted or divided from each other pairwise, until a collection of simulated difference spectra combining the variability inherent to each measurement is produced. From these bootstrapped simulations, confidence bands can be produced using any of the methods in Section 4.
Figure 6a and 6b show the ratio and difference between the IN spectra of water aged volcanic ash and unaged volcanic ash with confidence intervals. Suspension of minerals and volcanic ash in water can cause alteration of the ice-active surface sites due to a variety of geochemical processes as shown in recent literature (Harrison et al., 2019; Jahn et al., 2019; Kumar et al., 2019; Maters et al., 2020; Perkins et al., 2020; Fahy et al., 2022b). Based on the confidence bands of either metric, it can easily be seen that below approximately –12 °C, there is a statistically significant difference between the IN activity of the
aged ash and unaged ash with $p<0.05$, confirming that in this experiment there is an alteration of the IN activity of volcanic ash due to suspension in water. The magnitude of this difference has also been determined in both relative and absolute terms, providing a quantitative measurement of the change in IN activity due to chemical processing of this sample. In this case, the

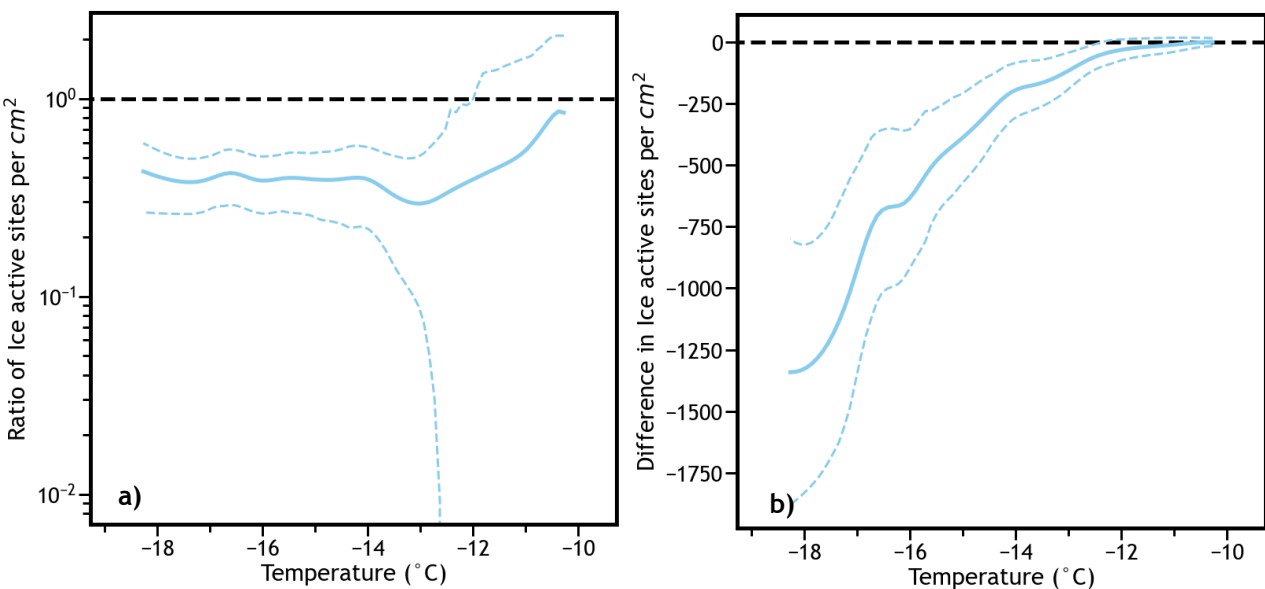

**Figure 6.** Comparison of the water aged $n_s$ spectrum to the unaged $n_s$ spectrum a) by dividing, and b) by subtracting. The dotted line appears at $\Delta n_s = 1$ in a)and at $\Delta n_s = 0$ in b), signifying no difference between the two spectra. Confidence bands are calculated using interpolated tskew bootstrapped ratio/difference simulations as discussed above. Based on the confidence bands, there is a statistically significant difference between the water aged and unaged ash spectra for temperatures $< -12$ ºC for both plots, and the magnitude of this difference is available as a function of temperature.

IN activity of water aged ash is reduced by a factor of 2-3 below –12 °C and is reduced by between 0 and 1500 ice active sites per square centimeter of ash surface area as a function of temperature. For this analysis only about 200 bootstrap simulations per shared degree Celsius seems to be necessary for consistent confidence bands for each difference spectrum.

**5.2 Applications of difference spectra**

Difference spectra have a variety of useful applications within the context of ice nucleation. The first has already been shown, as two spectra can be easily tested to determine whether there is a statistically significant difference between them. This is particularly useful in studies of chemical aging, where the change in IN activity after a given chemical treatment can be quantitatively measured using the difference or ratio before and after aging. Another application is in background freezing subtraction for IN spectra. All droplet-on-substrate methods used to measure heterogeneous IN activity have some level of background freezing activity either from background heterogeneous nucleation or from homogeneous ice nucleation that can change day-to-day depending on the system (Polen et al., 2018; Vali, 2019). For accurate measurements and to compare between instruments, the instrumental background (or homogeneous ice nucleation activity) must be subtracted from any measured heterogeneous IN spectrum. This can be readily accomplished by calculating the difference between the IN spectrum of interest and the background freezing spectrum. Where there is no background, the difference is equal to the sample spectrum. By saving the subtracted simulations used to calculate the variability in this difference spectrum, the background-subtracted

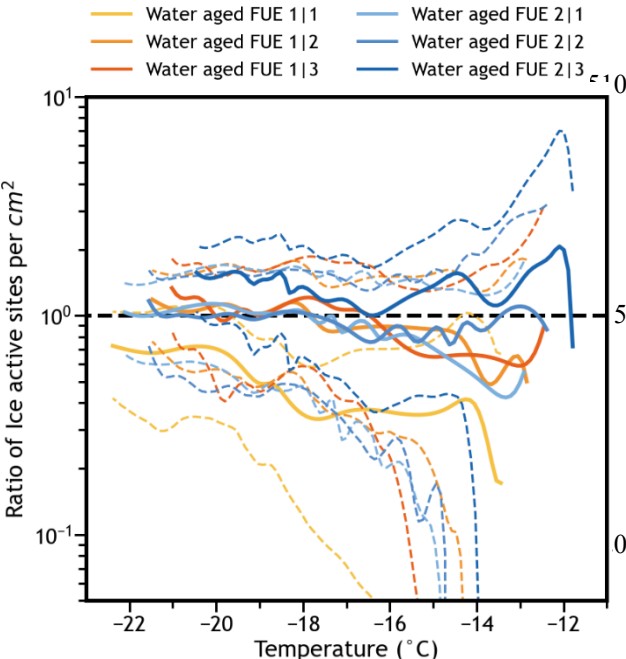

**Figure 7.** Comparison of individual water aged cumulative $n_s$ spectra to the remaining combined water aged cumulative $n_s$ spectra. Each curve is a difference spectrum of one of the six experiments divided by the remaining five experiments combined (solid lines) with bootstrapped tskew 99% confidence intervals (dotted lines). The experiment shown in goldenrod (1|1) is statistically significantly different from the other five experiments and is therefore deemed an outlier.

data can be compared further via another difference spectrum if desired. This can also be useful in determining whether a sample's IN activity is distinguishable from the instrumental background in weak IN materials. For all use cases, accurate confidence bands based on the bootstrapping procedures presented here are integral to ensuring rigorous and correct analysis and interpretation of the data, as simply subtracting $K(T)$ or $k(T)$ without accurate confidence intervals or other statistics does not fully represent the background-subtracted spectrum.

A third application of difference spectra in IN activity is in locating outliers. Droplet-on-substrate IN measurements are extremely sensitive to contamination and human error, even when great care is taken during the sample preparation process. When two measurements of the same sample disagree, additional replicate measurements are taken to determine if a measurement is an outlier, usually visually. Ideally, a more quantitative measurement of outlier status would be used, such as the Grubbs Test (Grubbs, 1969), Tukey's Fences (Tukey, 1977), or the Modified Thomson Tau Test (Thompson, 1985). However, the usefulness of these

common techniques and the assumptions they require for IN spectra is questionable. Instead, we propose that for a quantitative measurement of whether a sample is an outlier, the difference spectrum comparing the sample in question with the combined spectrum of the remaining measurements of the same sample can be used. An example of this analysis is shown in Figure 7, where the various water aged ash freezing experiments are compared using a difference plot to combinations of the remaining measurements. It can be clearly seen that only the spectrum shown in purple is statistically significantly different (in this case lower) at the 99% confidence level based on the bootstrapped tskew confidence bands. Therefore, this experiment could be treated as an outlier at that confidence level and excluded from future analysis. Even still, great care should be taken when dealing with potential outliers, and the confidence level required to exclude outliers should be carefully considered so as not to remove valid data. Whenever possible, decisions about whether to exclude a potential outlier should combine this statistical method with observations or lack thereof of specific experimental errors in the laboratory.

## 6 Summary and Conclusions

We have presented a rigorous and generalized set of methods for interpolating raw data, calculating confidence bands and other statistics, and quantitatively comparing IN spectra derived from droplet freezing assays. The interpolation methods discussed use ice nucleation data far more efficiently than previous binning methods and allow continuous quantitative comparison of IN spectra without compromising statistical power and detail present in the original data. Empirical bootstrapping is introduced as an improvement on the elementary statistical methods and parametric bootstrapping previously

used by capturing the full variability present in each IN spectrum or collection of IN spectra with no assumptions about the nature of ice nucleation for the material being tested. Enhanced continuous confidence bands are calculated using rigorous and modern algorithms to replace the quantile intervals or z-intervals previously used. Finally, the ability to interpolate and simulate IN spectra is used to develop difference spectra with accurate confidence bands for quantitative comparison and statistical testing of ice nucleation activities between materials and background subtraction.

These approaches can be used to help answer many important research questions in the field related to statistically assessing observed changes or differences in IN activities and can be applied to any experimental setup using arrays of droplets freezing over time or at varying temperatures. They are supported by statistical theory and use widely accepted methodologies from the statistics literature. The universality, simplicity, and accuracy of this approach makes it an ideal candidate to be a standard statistical method by which to compare datasets from different instruments and groups. The bootstrapping approach

could be particularly useful for incorporating uncertainty in IN activity into advanced atmospheric models, as a full distribution of IN activity at each temperature can be easily estimated from simulations. To facilitate adoption of these statistics, all code developed for this project along with documentation and data to recreate the figures in this paper is available in archived form as was used at the time of writing at KiltHub (Fahy et al., 2022a) or in a living GitHub repository where updates or additional information may be added in the future (https://github.com/wdfahy/CMU-INstats).

Further refinement of these methods by optimizing code runtime, improving confidence interval coverage, adding simulation methods, and implementing different statistics may be accomplished in the future as necessary. Extension of the procedures described here may be possible to describe uncertainty in instruments that measure ice nucleation in the aerosol phase such as CFDC-type instruments and expansion chambers and are not limited to ice nucleation. This may lead to applications describing uncertainty in experiments analyzing a variety of nucleation processes under varying conditions. If

widely adopted, the quality and consistency of statistical treatment of nucleation data will improve, leading to enhanced representation and communication of results and interpretations within those fields.

## Code availability

All code used in this project can be accessed in its archived form at: doi:10.1184/R1/19494188, with any updates or further work posted to https://github.com/wdfahy/CMU-INstats.

## Data availability

All data used in this project can be accessed at doi:10.1184/R1/19494188.

## Supplement link

See additional document.

## Author Contribution

WDF and RCS conceptualized the paper. CRS contributed statistical methods. WDF wrote the script; collected, analyzed, and visualized data, and wrote the initial manuscript draft. All authors provided input into the methods developed and edited the manuscript.

## Competing interests

There are no competing interests.

## Acknowledgements

This research was funded by National Science Foundation of the United States of America (CHM-1554941). We are grateful to Leif Jahn for helpful discussions in developing this concept and for two anonymous reviewers for their feedback which has greatly improved the clarity of this manuscript.

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
