# Peer review of "A universally applicable method of calculating confidence bands for ice nucleation spectra derived from droplet freezing experiments"

_Atmospheric Measurement Techniques, 2022_

## Referee Comment (RC1)

The manuscript titled "A universally applicable method of calculating confidence bands for ice nucleation spectra derived from droplet freezing experiments" by Fahey et al., demonstrates a method to derive ice nucleation (IN) spectrum and confidence bands for ice nucleation studies. They use this to determine an IN spectrum normalized to particles surface area, a quantity referred to as $n_s(T)$, the number of ice nucleation events normalized to particle surface area as a function of temperature. Bootstrapping is used in this method to simulate freezing temperatures multiple times by random sampling and calculating $n_s(T)$ from these simulated data. Variability in simulated freezing and $n_s(T)$ is used to derive confidence bands. Finally, the authors are then able to deduce statistical similarity between datasets. Overall, this paper stresses in the importance of deriving uncertainties in $n_s(T)$ and should be of great benefit to ice nucleation community.

Unfortunately, there are many instances when clarity and precision are lacking in describing the methods, as well as when the authors mix what can be stated as options or as claims with evidence. I cannot recommend publication until the comments below are addressed. The first major comment relates to poor writing, due to poorly explaining (or not explaining at all) complex topics/methods that are the main subjects of this manuscript. Another major comment is about assuming variability on the droplet by droplet basis in Eq 1 and 2. A final major comment is about the method for assessing the similarity of IN spectra is not explained at all. Although, the paper is quite novel, addresses relevant scientific questions, and has important conclusions. They demonstrate sufficient evidence through their simulations to support their conclusion.

Major Comments:
1. The article and especially the introduction has many vague terms, descriptions and opinions. I have outlined most in the minor comments below, however, a few major comments are described. Most notably the introduction is poorly written, with insufficient details about what the author will present and how it relates to previous work. The authors have not reviewed previous literature and what advances were made in ice nucleation simulation to lead them to this work. They are not the first to model or simulate ice nucleation. Instead, their intro is a narrative about the failure in the community. This is unacceptable and must be rewritten.
   a. "ice nucleation processes" The authors must be more specific about what this process is. As a reader, the phrase "it is simply a fact of the ice nucleation process" is meaningless because these facts are not outlined or given in the manuscript up to this point. I read only that ice nucleation is a process, and it does not help understanding. Please be specific, what is the process.
   b. "IN specta", "$n_s(T)$", "k", "K", These are defined in many places throughout the first pages of the manuscript. Would the authors please consolidate a common and non-redudant definitions of these.
      i. p. 3, l. 75: k and K are ice active site density spectra
      ii. p. 3, l. 76: k is a differential spectra and K is a cumulative spectra
      iii. p. 4, l. 81: K is the number of ice active sites at temperature T per unit of suspension volume. K can also be nm. K can also be ns.
      iv. p. 4, l. 6-7: K are the cumulative ice nucleation active site density curves
   c. p. 2, l. 44-46, This sentence insults the ice nucleation community. Yes, ice nucleation analysis is complex, however, the statistical tests, approximations, and methods of previous work are not inaccurate. They are peer-reviewed and explained. Please rewrite this sentence, emphasize previous work that has been done on which your manuscript is based, and how the authors fill a missing gap.
   d. p. 4, l. 81: "…number of ice active sites…" This is a vague term. What is a site and how does Eq 1 and 2 give sites? In fact, eq1 and eq2 are using counted freezing events and

there are no ice active sites in them. There are many assumptions made between Eq 1 and 2, and then making a claim about quantitative measures of ice active sites.

    e. The word "complex" is overused. On p. 5, l. 23, what make a curve complex and what makes a curve not complex? On p. 2, l. 55, what makes a dataset complex and not complex? On p. 2, l. 45, what makes ice nucleation complex and what makes it not complex? On p. 11, l. 80, math is complex. It appears that when the authors do not want to take the time to explain ice nucleation, IN specta/curves on graphs, and bootstrapping, they call it "complex" and move on. This shows lack of care and work put in to write a comprehensive manuscript, as these complex things are the main subjects of the manuscript. Please search for the word complex in the manuscript and try to replace it with specific details and explanation. These should be briefly explained.

    f. The words/phrases "superior", "best", "most powerful" are used through the manuscript. In all context, these are opinions of the authors. I suggest removing all.

2. Equation 1 and 2 has an important and unclaimed assumption. Why are the authors so sure that the normalization constant, X, is identical for each droplet? On l. 35-37, it states many differences that could cause variability from droplet to droplet. If so, these equations are not valid. If X were to be different for each droplet, then it would need to be accounted for. However, this equation inherently assumes some identical normalization. In other words, one constant for all liquid droplets. In other derivations besides Vali, such as in the Pruppacher and Klett textbook, eqs 7-64 and 9-56, the natural log appears due to the assumption that all drop volumes are the same, or surface area in drops is the same. The authors must claim this assumption in the manuscript, and that their analysis only holds if this assumption is valid.

3. p. 15, l. 92 - p. 18, l. 97: This method descriptions is far too short. The authors claim taking a ratio or subtracting two different $n_s(T)$ curves is supposed to be between 0 and 1? That is the test of similarity. When taking a ratio or difference, the authors are doing this on a log10 scale? When they take the ratio, does the larger spectra need to be in the denominator? When taking the difference, the smaller values need to be subtracted from the larger values to get only positive numbers? The step-by-step procedure here is not so clear and so it is difficult to review Fig. 5 and any discussion around it. It would help to guide the reader a bit more slowly here. Finally, can the authors give a name or equations for "these metrics" stated p. 15 l. 91?

Minor Comments:

1. Abstract, p. 1, l. 15: "…and if used properly…" Would you please rephrase this? It reads as if there is a way to also improperly use your methods.

2. p. 1, l. 19: "By improving the statistical tools available, this work will improve…" Can you rephrase this? It reads like the authors still need to improve their methods. I do not think that is their intension.

3. p. 1, l. 19: "…improve the quality and accuracy…" Accuracy is improved with instrumentation that is more accuracy. Statistical analysis does not make instruments more or less accurate. Would you please state exactly what is more accurate here?

4. p. 1, l. 39: The authors write the word "this", but do they refer to all of the uncertainties and variabilities in experimental investigation mentioned above? Or are they still talking about a perfect experimental setup?

5. Intro, p. 2, l. 33: Would the authors please precisely define "IN activity"? There should be a distinction between what is measured and what is derived.

6. p. 2, l. 42: "If we cannot eliminate experimental error, it must…" The word "we" is usually referred to the authors. I suggest to replace with "Experimental error is always present and must be …"

7. p. 2, l. 43-44: It is the authors opinion that there is no widely accepted approach. Please remove this sentence, or write this is an opinion.

8. p. 2, l. 50: "…remaining questions…" What are the questions the author is referring? What are those questions that remain? Please be precise in formulating your scientific questions.

9. p. 2, l. 50: "…experimental parameters…" What are the specific experimental parameters the authors are referring to. Please be specific.

10. p. 2, l. 51: "…these approaches …" What are the approaches the authors are referring to. Please be specific.

11. p. 2, l. 55: What is the difference between a toy IN dataset and an IN dataset. Maybe the authors would like to change this to read, "example of an IN dataset"?

12. p. 2, l. 61: please state the diameter

13. p. 3, l. 68-73: These sentences are redundant. Please rewrite.

14. p. 4, l. 99: Should "…temperature density of freezing events…" be "…number of freezing events depending on temperature…"?

15. p. 5, l. 49-40: This is a strange way to end. This is a strange way to end. Please tell the reader exactly what you are looking for, instead of telling them you are looking "elsewhere".

16. p. 6, l. 44 and l. 49: What is the difference between the terms "accurate interpolation", "faithful interpolation", and simply "interpolation". Can an interpolation be "unfaithful" or "inaccurate"? I can find no quantitative meaning of these in the paper. I suggest to remove "accurate" and "faithful". Please search through the manuscript for these.

17. p. 8, l. 83: "…are unreliable." should be changed to "…may be unreliable."

18. p. 8, l. 88-89: I tried looking up the reference Kaufmann et al. 2017, but it is not in the references list at the end of the manuscript. Would you please check that all references are actually included?

19. p. 10, l. 45: "…originally proposed…" What exactly was originally proposed? There are many things proposed up until this point reading the manuscript.

20. p. 10, l. 47: …"with replacement…" Would the authors please be specific about what this means? I thought it was a typo at first, but appears like it means something important to them.

21. p. 10, l. 61: "bye" is a typo.

22. p. 10, l. 62-65: This sentence is unnecessary, negative and offensive. Please remove it. It is affronting members of the ice nucleation community as unfamiliar with statistics and as unable to access information. This is not true.

23. Through the manuscript there is use of single quoted words and phrases. It is not clear to the reader why these have to have single quotes. Would the authors please elucidate the reason for this, or just remove the quotes and explain things clearly.

24. p. 11, l. 78: What is a "re-interpolation", and how can this be accurate or inaccurate?

25. p. 12, l. 4: Please write out what alpha is.

26. p. 15, l. 82: Please add commas so that the words "…method such as empirical bootstrapping rooted…" is changed to "…method, such as empirical bootstrapping, rooted…".

27. p. 17, l. 22-24: This is plenty of evidence that droplets on a substrate surrounded by oil or air can be used to measure homogeneous freezing. Yes, other studies have a background of heterogeneous ice nucleation occurring. Generalizing this to all substrate based approaches is not correct. Please remove this sentence.

---

## Author Response (AR1)

**Response to Referee #1**

The manuscript titled "A universally applicable method of calculating confidence bands for ice nucleation spectra derived from droplet freezing experiments" by Fahy et al., demonstrates a method to derive ice nucleation (IN) spectrum and confidence bands for ice nucleation studies. They use this to determine an IN spectrum normalized to particles surface area, a quantity referred to as ns(T), the number of ice nucleation events normalized to particle surface area as a function of temperature. Bootstrapping is used in this method to simulate freezing temperatures multiple times by random sampling and calculating ns(T) from these simulated data. Variability in simulated freezing and ns(T) is used to derive confidence bands. Finally, the authors are then able to deduce statistical similarity between datasets. Overall, this paper stresses in the importance of deriving uncertainties in ns(T) and should be of great benefit to ice nucleation community.

Unfortunately, there are many instances when clarity and precision are lacking in describing the methods, as well as when the authors mix what can be stated as options or as claims with evidence. I cannot recommend publication until the comments below are addressed. The first major comment relates to poor writing, due to poorly explaining (or not explaining at all) complex topics/methods that are the main subjects of this manuscript. Another major comment is about assuming variability on the droplet by droplet basis in Eq 1 and 2. A final major comment is about the method for assessing the similarity of IN spectra is not explained at all. Although, the paper is quite novel, addresses relevant scientific questions, and has important conclusions. They demonstrate sufficient evidence through their simulations to support their conclusion.

**We thank the referee for their detailed review of our manuscript. Each comment made is responded to below, along with any revisions as appropriate.**

1. The article and especially the introduction has many vague terms, descriptions and opinions. I have outlined most in the minor comments below, however, a few major comments are described. Most notably the introduction is poorly written, with insufficient details about what the author will present and how it relates to previous work. The authors have not reviewed previous literature and what advances were made in ice nucleation simulation to lead them to this work. They are not the first to model or simulate ice nucleation. Instead, their intro is a narrative about the failure in the community. This is unacceptable and must be rewritten.

**Our original manuscript was formatted such that the review of previous literature and previous advances is contained mostly in later sections, where each specific technique is introduced and explained. To improve the introduction, some language summarizing previous work has been added, along with a sentence directing the reader to the later sections where this work is discussed in more detail.**

> **After p.2 l.46: "In our experience, there is no widely implemented approach to reporting uncertainty in IN temperature spectra derived from freezing experiments. Instead, methods vary between groups, often relying on different assumptions about the nature of ice nucleation measurements, the forms of distributions that the random variables involved take, and the quantification of the derived uncertainties. In the simplest case, standard deviations, errors, and/or confidence intervals have been calculated from repeated experiments either by assuming that variability follows a normal distribution (Polen et al., 2018; Jahn et al., 2019; Worthy et al., 2021; Chong et al., 2021; Roy et al., 2021; Losey et al., 2018), a Poisson distribution, Koop et al. (1997) (Yun et al., 2021; Alpert and Knopf, 2016; Knopf et al., 2020; Kaufmann et al., 2017), or that droplet freezing follows a binomial**

distribution (Gong et al., 2020, 2019; McCluskey et al., 2018; Suski et al., 2018; Wex et al., 2019). In other cases, authors have used a model of ice nucleation to simulate their experiments and use that simulated distribution to estimate the uncertainty present in their experiment. In the simplest case, droplet freezing is modelled as a Poisson point process (Vali, 2019; Fahy et al., 2022b; Jahl et al., 2021). In more sophisticated models, variables such as the number of sites, mass of material, and temperature variations are parameterized to run completely new simulated experiments (Wright and Petters, 2013; Harrison et al., 2016). Even in these sophisticated models, either additional measurements are required, or assumptions must be made about the distribution of each variable. Until the inherent variability behind ice nucleation can be measured to prove or disprove the assumptions being made, all the above methods are only as reliable as the assumptions themselves. In Section 4, each method, their required assumptions, and the validity of those assumptions are discussed in detail.

Empirical bootstrapping is an alternative approach to estimating statistics for a dataset that to our knowledge has not been applied in the context of ice nucleation. In this technique, a series of random samples of the measured dataset is taken to generate estimated statistics that converge on the actual values as the number of samples increases (Efron, 1979; Shalizi, forthcoming). No assumptions are required about the distributions of random variables underlying ice nucleation and it can be applied to any system where the freezing temperatures or times of droplets are measured. Here we present a set of generalized and statistically rigorous methods based on empirical bootstrapping for quantifying uncertainty in IN spectra. When accompanied by interpolation methods presented in Section 3, this approach can be used to calculate continuous confidence bands and statistically test differences between IN spectra (Section 5). We also address the effects of interpolation techniques, droplet sample size, and bootstrap sample size to direct the field towards more rigorous and repeatable methods of experimentation and data analysis. An implementation of all presented statistical methods along with documentation and instructions for its use is provided freely for use or reference to assist in future research and improve the statistical treatment of ice nucleation data in the field."

a. "ice nucleation processes" The authors must be more specific about what this process is. As a reader, the phrase "it is simply a fact of the ice nucleation process" is meaningless because these facts are not outlined or given in the manuscript up to this point. I read only that ice nucleation is a process, and it does not help understanding. Please be specific, what is the process.

**Changed this sentence to:**

"This variability is inherent to ice nucleation. Using the singular-stochastic model most recently discussed in Vali (2014) and terminology proposed in Vali and DeMott et al., (2015), ice nucleation activity (or rate) is an accumulation of many ice nucleation sites with variable critical active temperatures dispersed randomly throughout a material. In turn, the material is distributed randomly throughout droplets which can have varying sizes, shapes, and environments. Therefore, a measured IN activity can be affected by heterogeneity in the distribution of ice active sites across a material, heterogeneity in the mass or surface area of material suspended in each droplet, differences between droplet sizes and environments, and variations in temperature between droplets.

b. "IN specta", "ns(T)", "k", "K", These are defined in many places throughout the first pages of the manuscript. Would the authors please consolidate a common and nonredudant definitions of these. i. p. 3, l. 75: k and K are ice active site density spectra ii. p. 3, l. 76: k is a differential spectra and K is a cumulative spectra iii. p. 4, l. 81: K is the number of ice active sites at temperature T per unit of suspension volume. K can also be nm. K can also be ns. iv. p. 4, l. 6-7: K are the cumulative ice nucleation active site density curves

**Two of these definitions have been removed ("k and K", p. 3, l. 75 and "For simplicity and generality… cumulative or differential IN spectrum", p. 4 l. 85-86). The remaining instances either define the terms themselves or the connection between k/K and other terms.**

c. p. 2, l. 44-46, This sentence insults the ice nucleation community. Yes, ice nucleation analysis is complex, however, the statistical tests, approximations, and methods of previous work are not inaccurate. They are peer-reviewed and explained. Please rewrite this sentence, emphasize previous work that has been done on which your manuscript is based, and how the authors fill a missing gap.

**It was never our intention to insult the ice nucleation community, and we apologize that it was written to imply that. Indeed, the previous statistical methods used are peer-reviewed and explained. However, as we discuss in detail later in the manuscript, their accuracy varies depending on the validity of the assumptions and approximations made to derive the statistics used. This is not a failing of the ice nucleation community – every statistic has an error associated with it, and for reasons discussed elsewhere in the manuscript, ice nucleation presents a difficult problem for many standard statistical methods. We merely wished to point out that the approaches being used for some studies (including our own previous works) may not have been as accurate as many people reading them may have assumed.**

**To remove any offensive language, this sentence has been changed to:**

> **"In our experience, there is no widely implemented approach to reporting uncertainty in IN temperature spectra derived from freezing experiments. Instead, methods vary between groups, often relying on very different assumptions about the nature of ice nucleation, the form of distribution which the random variables involved take, and the visualization of the derived uncertainties."**

d. p. 4, l. 81: "…number of ice active sites…" This is a vague term. What is a site and how does Eq 1 and 2 give sites? In fact, eq1 and eq2 are using counted freezing events and there are no ice active sites in them. There are many assumptions made between Eq 1 and 2, and then making a claim about quantitative measures of ice active sites.

**Much of this material is thoroughly discussed in literature that we cite in this paper. We have added some additional discussion of what ice nucleation sites are and the model we are using to conceptualize ice nucleation earlier in the text to point the reader in the direction of that literature.**

e. The word "complex" is overused. On p. 5, l. 23, what make a curve complex and what makes a curve not complex? On p. 2, l. 55, what makes a dataset complex and not complex? On p. 2, l. 45, what makes ice nucleation complex and what makes it not complex? On p. 11, l. 80, math is complex. It appears that when the authors do not want to take the time to explain ice nucleation, IN specta/curves on graphs, and bootstrapping, they call it "complex" and move on. This shows lack of care and work put in to write a comprehensive manuscript, as these complex things are the main subjects of the manuscript. Please search

for the word complex in the manuscript and try to replace it with specific details and explanation. These should be briefly explained.

**Most examples of 'complex' have been removed, we agree it was overused. However, in some cases the word complex is used rightfully – in the case of what makes ice nucleation complex, there is an extensive body of previous literature explaining the complexity. We have incorporated a brief discussion the model that we are referencing, but most of that material is the subject of a different manuscript. Similarly, for derivations of IN spectra, k and K are widely used and accepted in the community, and have already been explained and discussed in detail elsewhere – redoing those calculations and explanations is outside of the scope of this manuscript.**

**We have added some citations about the mathematics behind bootstrapping, but once again, this manuscript is not intended to prove the efficacy of bootstrapping in general – it is an application of a well-defined and previously studied tool in statistics. The new parenthetical statement is:**

> **"(see Efron and Tibshirani, (1994) or Davison and Hinkley, (1997) for a thorough treatment of the mathematics behind bootstrapping and Canty et al., (2006) for a thorough discussion of inconsistencies and errors that can be encountered when using bootstrapping),"**

f. The words/phrases "superior", "best", "most powerful" are used through the manuscript. In all context, these are opinions of the authors. I suggest removing all.

**These words have been removed, except for in the following case:**

**p. 14, l. 344, 'best-performing' remains because we define a metric for what we are interpreting that to mean, and studentized bands are known to be the most accurate confidence bands that we used.**

2. Equation 1 and 2 has an important and unclaimed assumption. Why are the authors so sure that the normalization constant, X, is identical for each droplet? On l. 35-37, it states many differences that could cause variability from droplet to droplet. If so, these equations are not valid. If X were to be different for each droplet, then it would need to be accounted for. However, this equation inherently assumes some identical normalization. In other words, one constant for all liquid droplets. In other derivations besides Vali, such as in the Pruppacher and Klett textbook, eqs 7-64 and 9-56, the natural log appears due to the assumption that all drop volumes are the same, or surface area in drops is the same. The authors must claim this assumption in the manuscript, and that their analysis only holds if this assumption is valid.

**We have added the following language to explicitly claim this assumption in deriving IN active site density spectra. You are correct that the calculation of these spectra only holds if this assumption is valid, as we now make explicit:**

> **"The derivation of these equations requires that $X$ be identical for every droplet being analyzed – an important assumption and source of error."**

**We have also made explicit the connection between this normalization constant and our statistical methods – specifically, if empirical bootstrapping is used, then variations in the normalization constant X are automatically incorporated into the error analysis:**

> **"However, as will be discussed later, the empirical bootstrapping approach quantifies this source of error, meaning these parameters can be used and interpreted even when the assumption does not strictly apply as long as the uncertainty is also incorporated into the interpretation."**

**And later on in Section 4:**

> **"Since variations in droplet size, sample mass suspended, or distributions of surface area among droplets (the parameters behind the normalization constant $X$) also contribute to the variability observed in experiments, the error caused by assuming $X$ is constant between droplets is also included into the model."**

3. p. 15, l. 92 - p. 18, l. 97: This method descriptions is far too short. The authors claim taking a ratio or subtracting two different ns(T) curves is supposed to be between 0 and 1? That is the test of similarity. When taking a ratio or difference, the authors are doing this on a log10 scale? When they take the ratio, does the larger spectra need to be in the denominator? When taking the difference, the smaller values need to be subtracted from the larger values to get only positive numbers? The step-by-step procedure here is not so clear and so it is difficult to review Fig. 5 and any discussion around it. It would help to guide the reader a bit more slowly here. Finally, can the authors give a name or equations for "these metrics" stated p. 15 l. 91?

> **We have added some additional detail. To clarify, all we are doing is subtracting or dividing two spectra and calculating resulting confidence bands – there is not much more information to provide beyond how confidence intervals are calculated, which is also explicitly stated. To answer your questions, we are not working on a log10 scale except to plot the data clearly. No, the larger spectrum does not need to be in the denominator, nor does the smaller value have to be subtracted from the larger value – note that in Figure 5, neither of these conditions are true. We do not claim that the ratio or (mathematical) difference between two ns spectra is 'supposed' to be between 0 and 1 – we are discussing the confidence bands of each metric and making conclusions about whether two IN spectra are different based on whether they contain 0 or 1.**

> **We have clarified the language. We use the term 'difference spectrum' to represent both metrics we are discussing to make this section less verbose. The paragraph now reads:**

> > **"For a more quantitative (and interpretable) method to compare IN spectra can simply be divided or subtracted. We will use the term 'difference spectrum' to refer to a continuous plot of either this ratio or difference, as both are calculated using the same procedures and provide similar information. When interpolated IN activity spectra are used, a continuous difference spectrum can easily be generated by calculating the ratio (or difference) between two interpolations at each point in a dense grid of temperatures, then interpolating between those points. A difference spectrum can be plotted as a function of temperature with its own confidence bands and can be used to test whether two IN spectra are statistically significantly different at any temperature where the two spectra overlap at any confidence level. Stated precisely, in the case of a ratio-based difference plot with confidence bands, the hypothesis that the two IN spectra are different can be tested. If the confidence bands do not contain one at a given temperature, then the null hypothesis (that the two IN activities compared are insignificantly different at that temperature) is rejected. If they do contain one, then that claim cannot be made. If a difference between IN spectra is used instead of a ratio, then zero is used for this hypothesis test instead of one. Therefore, if confidence bands can be accurately calculated for a difference spectrum, then continuous statistically rigorous claims about differences between IN spectra can be tested."**

> **Some additional language in later paragraphs has also been changed to match the terminology used here.**

1. Abstract, p. 1, l. 15: "…and if used properly…" Would you please rephrase this? It reads as if there is a way to also improperly use your methods.

**This phrase has been changed to:**

> **"…and when large sample sizes (~>150 droplets and >= 1000 bootstrap samples in our system) can capture…"**

2. p. 1, l. 19: "By improving the statistical tools available, this work will improve…" Can you rephrase this? It reads like the authors still need to improve their methods. I do not think that is their intension.

**This has been rephased to:**

> **"By providing additional statistical tools to the community, this work will improve…"**

3. p. 1, l. 19: "…improve the quality and accuracy…" Accuracy is improved with instrumentation that is more accuracy. Statistical analysis does not make instruments more or less accurate. Would you please state exactly what is more accurate here?

**This sentence now reads:**

**"improve the quality and accuracy of statistical tests and uncertainties in…"**

4. p. 1, l. 39: The authors write the word "this", but do they refer to all of the uncertainties and variabilities in experimental investigation mentioned above? Or are they still talking about a perfect experimental setup?

**Changed to "This variability"**

5. Intro, p. 2, l. 33: Would the authors please precisely define "IN activity"? There should be a distinction between what is measured and what is derived.

**Inserted the parenthetical statement:**

> **"(here we use the term 'IN activity' as a general term to refer to any measured or derived variable which quantifies ice nucleation rate with respect to temperature.)"**

**In this case we do not make a distinction between measured and derived values because they are different quantifiers of the same underlying process, and as such have the same sources of error. Later in the manuscript we specify which variables are measured and which are derived in the framework we are using.**

6. p. 2, l. 42: "If we cannot eliminate experimental error, it must…" The word "we" is usually referred to the authors. I suggest to replace with "Experimental error is always present and must be ..."

**This change has been made.**

7. p. 2, l. 43-44: It is the authors opinion that there is no widely accepted approach. Please remove this sentence, or write this is an opinion.

**This has been changed to "In our experience there is no…"**

8. p. 2, l. 50: "…remaining questions…" What are the questions the author is referring? What are those questions that remain? Please be precise in formulating your scientific questions.

9. p. 2, l. 50: "…experimental parameters…" What are the specific experimental parameters the authors are referring to. Please be specific.

**To replace both of these statements this line has been changed to:**

> **"We also address the effects of interpolation and sample size on measured IN activity spectra."**

10. p. 2, l. 51: "…these approaches …" What are the approaches the authors are referring to. Please be specific.

**Changed to "all presented statistical methods" and added "based on empirical bootstrapping" to a previous sentence for more specificity.**

11. p. 2, l. 55: What is the difference between a toy IN dataset and an IN dataset. Maybe the authors would like to change this to read, "example of an IN dataset"?

**This now reads "we selected an example IN dataset".**

12. p. 2, l. 61: please state the diameter

**Added in text, "(1.5 mm)."**

13. p. 3, l. 68-73: These sentences are redundant. Please rewrite.

**The second to last sentence has been removed ("The validity of… suspensions.").**

14. p. 4, l. 99: Should "…temperature density of freezing events…" be "…number of freezing events depending on temperature…"?

**Phrasing has been changed to "number density of freezing events with respect to temperature".**

15. p. 5, l. 49-40: This is a strange way to end. This is a strange way to end. Please tell the reader exactly what you are looking for, instead of telling them you are looking "elsewhere".

**Changed to "we must look for an interpolation method that can capture an ice nucleation spectrum with any shape"**

16. p. 6, l. 44 and l. 49: What is the difference between the terms "accurate interpolation", "faithful interpolation", and simply "interpolation". Can an interpolation be "unfaithful" or "inaccurate"? I can find no quantitative meaning of these in the paper. I suggest to remove "accurate" and "faithful". Please search through the manuscript for these.

**When referring to interpolations, these terms have been removed in most cases. However, the term 'accurate' is used in many places in the manuscript to indicate that a particular approach either qualitatively matches the shape and value of the 'actual' values (either measured or theoretical). When referring to confidence limits, for example, there does exist an actual exact confidence limit at every point, even though we cannot measure it without knowledge of the entire population. Thus, it is still valid (and informative) to use the term 'accurate' in relative terms between methods.**

17. p. 8, l. 83: "…are unreliable." should be changed to "…may be unreliable."

**Changed the wording to "distribution, making this assumption unreliable."**

18. p. 8, l. 88-89: I tried looking up the reference Kaufmann et al. 2017, but it is not in the references list at the end of the manuscript. Would you please check that all references are actually included?

**We have double-checked the reference list to make sure nothing is missing. The requested citation is as follows:**

Kaufmann, L., Marcolli, C., Luo, B., & Peter, T. (2017). Refreeze experiments with water droplets containing different types of ice nuclei interpreted by classical nucleation theory. *Atmospheric Chemistry and Physics*, *17*(5), 3525–3552. https://doi.org/10.5194/acp-17-3525-2017

19. p. 10, l. 45: "…originally proposed…" What exactly was originally proposed? There are many things proposed up until this point reading the manuscript.

**Removed this parenthetical statement – empirical bootstrapping was proposed before any type of parametric bootstrapping, but it is not important for the manuscript, so we removed it to avoid confusion.**

20. p. 10, l. 47: …"with replacement…" Would the authors please be specific about what this means? I thought it was a typo at first, but appears like it means something important to them.

**Added the parenthetical statement "i.e. the same datapoint can be sampled more than once"**

21. p. 10, l. 61: "bye" is a typo.

**Thank you – changed to 'by'**

22. p. 10, l. 62-65: This sentence is unnecessary, negative and offensive. Please remove it. It is affronting members of the ice nucleation community as unfamiliar with statistics and as unable to access information. This is not true.

**We have removed this statement – we intended no offence, we simply wished to supply some rationalization for why this very simple technique that is well-suited for the problem of ice nucleation (empirical bootstrapping) has not been used previously.**

23. Through the manuscript there is use of single quoted words and phrases. It is not clear to the reader why these have to have single quotes. Would the authors please elucidate the reason for this, or just remove the quotes and explain things clearly.

**Single quotes are used to denote a term that we are defining. Much as you would quote someone else's term with double quotes, we are using single quotes to separate the specific term from the surrounding text as an important definition. For example, when comparing 'observed' vs 'sampled' lists, these words have single quotes to denote that we are defining these terms in this context. There are a few places where the single quotes were redundant, and those have been removed.**

24. p. 11, l. 78: What is a "re-interpolation", and how can this be accurate or inaccurate?

**Changed to "interpolated exactly using a simple spline fit" for clarity. We used re-interpolation to indicate that the curve had previously been interpolated, then discretized, and that we are now interpolating it again.**

25. p. 12, l. 4: Please write out what alpha is.

**We have added "respectively, where alpha is the threshold value chosen for statistical significance"**

26. p. 15, l. 82: Please add commas so that the words "…method such as empirical bootstrapping rooted…" is changed to "…method, such as empirical bootstrapping, rooted…".

**Changed.**

27. p. 17, l. 22-24: This is plenty of evidence that droplets on a substrate surrounded by oil or air can be used to measure homogeneous freezing. Yes, other studies have a background of heterogeneous ice nucleation occurring. Generalizing this to all substrate based approaches is not correct. Please remove this sentence.

**Changed the first word to "Most" instead of "All" and removed "measured as the ice active site density normalized to the volume of water in each droplet" for concision. However, background subtraction is important for many systems - even droplet-on-substrate methods that can measure homogeneous freezing usually have some heterogeneous freezing signal earlier in the spectrum and many systems that are intended to measure rare heterogeneous ice nucleation events have higher background freezing activities that should be accounted for. As such, we have elected to keep the sentence, but have removed the generalization.**

**Response to Referee #2**

This manuscript evaluates the potential of two types of bootstrapping methods for quantifying confidence intervals of droplet freezing experiments. Therefore, this manuscript may contribute to a more consistent interpretation of results from droplet freezing studies. The authors have also provided a very thorough documentation of their coding efforts through a publicly available Github repository.

**We thank the referee for their feedback on our manuscript. Each comment made is responded to below, along with any revisions as necessary.**

However, I have major concerns regarding the discussion of limitations and requirements associated with the bootstrapping methods evaluated in this paper. These aspects (e.g., sample size requirements) need to be discussed in much more detail, e.g., as in figure S3 which should be moved to the main text. A more detailed discussion would also ensure that these methods are used „properly" as mentioned in the abstract. I would also recommend a stronger focus on the bootstrapping methods, instead of an in-depth discussion of binning and interpolation methods, to make the manuscript more concise and less verbose

**In terms of limitations, besides the assumption of statistical independence stated in the manuscript there is only the issue of sample size/bootstrap sample size. For sample size requirements, we have brought what was Figure S3 into the main text and have also edited it – in the original figure, all curves included the freeze that was determined to be an outlier in Section 5.2, which caused the confidence bands on the N=98 and N=191 to be wider than they would normally be. Instead, we now randomly sample 50, 100, 150, and 200 droplets (along with the original 286) to**

better understand the effects of sample size on the width of confidence bands and the shape of the interpolations used.

Along with adding some additional text in Section 4.6, as you are right that it is important for any future implementation or interpretation of these methods. The last paragraph of Section 4.6 has been expanded significantly:

[revised manuscript text omitted]

We have also added a mention of these limitations/recommendations in both the last paragraph of the introduction and into the abstract to emphasize them.

However, we cannot generalize these requirements to every application of these techniques – the number of droplets and simulations required is dependent on the experimental setup, sample in question, and the questions the investigator is trying to answer. Each investigator will have to make their own choices about sample and resampling size depending on their findings. Figure S3 (now 5) is just an example of how this should be done and should not be taken as a general statement for the sample size requirements for empirical bootstrapping.

Regarding the in-depth discussion of interpolation/binning, interpolation is required for calculating continuous confidence bands and for comparing IN spectra quantitatively as discussed in Section 5, so we think it is integral to this manuscript. We are proposing to move away from the commonly-used binning techniques, so we offer a detailed explanation why and show the benefits of our preferred approach. However, we have moved Figure 2c and 2d to the SI, as most of the information in those panels is contained in panel a and b as well.

p.2, l. 58: HPLC is not defined - please make sure that abbreviations are defined consistently across the manuscript.

In this case, we are simply referring to a grade of chemical – we have changed it to the more conventional parenthetical “(HPLC grade, Sigma)”.

p.3, Fig. 1: The naming convention is slightly confusing - maybe remove duplicate mentioning of „unnamed FUE" and „water aged FUE" in the legend? Also, the color scheme is not color-blind friendly and differences between samples are hard to see. How many runs were included per sample?

**We have changed the legend to specify that the first two spectra are 'combined' spectra, while the rest are individual runs. We have also changed the color scheme and increased contrast (using the discrete rainbow scheme from Figure 19 of the following blog post: [https://personal.sron.nl/~pault/](https://personal.sron.nl/~pault/)), but our main point is not differentiating between individual runs, but rather showing that the combined spectra represent the individual runs relatively well. Unfortunately, it still is difficult to distinguish between all the spectra because by their very nature they overlap significantly – that's the point.**

p.3, l.75ff: I don't see the advantage of representing the droplet freezing data as ns values within the context of this study. However, if applied, the concept of ice-active site densities needs to be introduced explicitly - it is unclear how ns is derived and which specific surface area values are used.

**We chose to use ns values because that tends to be the most common normalization scheme in the ice nucleation community today. We have added an explicit equation describing the variables used to calculate ns and have added our source for the specific surface area we used. The concept of ice-active site densities is also introduced in the introduction in more detail.**

**We do not go into more detail on the subject in this paper because it is not the focus of this study, and many previous authors have discussed the concept of ice active site densities in detail.**

p.4, l.88f: What does „many" droplets mean exactly? Throughout the manuscript, a stronger emphasis should be on a discussion of the impacts that sampling size has (i.e., in the original observed data, the re-sampled data etc.)

**Changed to "hundreds of… in our estimation". In this case, we are basing this statement on previous work, but in other places we have emphasized the effects of sample size as mentioned in response to your major comment.**

p.5, l. 36: I would debate whether the contact angle approach with its many underlying assumptions (e.g., using bulk properties for describing microscopic nucleation processes) is strictly speaking „physically-based" - maybe rephrase?

**You are correct – we have simply removed "physically-based" – it is not necessary for the point we are making.**

p.6, l. 63: „high/warm" instead of „low" temperature?

**Changed, thank you for catching that.**

p.7, Fig.2d: Why does the interpolated data (unaged FUE) stop at -12 degC?

**Excellent question – in the version of the code used for this manuscript, a cutoff for the start of freezing events was used because without it, the interpolated spectra were unrealistic around the first few freezing events. We've reduced this cutoff so that more of the initial freezing events are captured, but we've also added the following line to explain this:**

"Note that the interpolated spectra do not start until there is a sufficient density of freezing events (more than one per degree Celsius) to avoid overfitting and because the error on these initial points is much larger than that of the rest of the spectrum as will be seen later."

p.8, l.75ff: Many of these approximations are only valid for „larger" droplet ensembles - this limitation should be mentioned to emphasize in which situations (i.e., for which experimental setups) we need more flexible statistical approaches. Also, as many readers might be more familiar with t-intervals than Z-intervals, a short explanation of the differences would be great.

For the Z- vs t-interval, the line has been changed to: "a Z-interval (based on the normal distribution) or t-interval (based on Student's t-distribution) can be constructed."

As for the main point of your comment here – the number of droplets required for an accurate closed-form confidence limit to be calculated is not straightforward.

For example, according to the NIST engineering statistics handbook section 7.2.2.2[1], for a two-sided t-test (a procedure essentially similar to constructing confidence intervals for a single process e.g. 7.2.2.1 of the NIST handbook), the sample size required is:

$$N = \left( t_{1-\frac{\alpha}{2}} + t_{1-\frac{\beta}{2}} \right)^2 (\frac{s}{\delta})^2$$

Where N is the sample size, $\alpha$ is the confidence level for the test, $\beta$ is the probability that we will fail to detect a shift of $\delta$ from the mean, and s is the standard deviation of the sample. We can set $\delta$ and $\beta$ depending on how accurate we want our test to be. Say we are willing to accept a 1% chance of missing (failing to reject the null hypothesis that the mean has shifted) a shift of 1 standard deviation for a test with 95% confidence. Then, the required sample size is:

$$N = \left( t_{1-\frac{0.05}{2}} + t_{1-\frac{0.01}{2}} \right)^2 (\frac{s}{1s})^2$$

If we assume that the number of degrees of freedom for our sample is equal to the number of droplets, in our system approximately 50 (a debatable assumption that will be discussed momentarily), then we get:

$$N = (2.009 + 2.678)^2 \approx 22$$

The question becomes, what does a sample size of 22 mean? If we are testing (or constructing a confidence interval) for the temperature at which the average droplet freezes, then this means 22 droplets are required from a single sample. However, what if we want to test the difference in frozen fraction or IN active site density spectrum at a specific temperature or the difference in temperature at which a given frozen fraction of IN active site density occurs (a much more common and useful set of questions)? Then, the most common approach when using simple t- or Z-intervals is to re-run the experiment several times, interpolate in some way, and then use the mean and standard deviation at a specific temperature or IN activity measurement to calculate the
* * *
[1] **NIST/SEMATECH e-Handbook of Statistical Methods, http://www.itl.nist.gov/div898/handbook/, accessed August 16th, 2022**

confidence intervals or perform the test (e.g. Polen et al., 2018[2]). In this case, is the sample size the number of replicate runs (e.g. the three separate freezing assays performed on the same sample), or the total number of droplets in all three of those replicate assays? Strictly speaking, since we are only getting a single parameter out of each of the three runs, we must assume the former, and therefore for the Central Limit Theorem to drive the distribution of these parameters to a normal or t-distribution where the t- or Z-interval is valid, we need many more replicate runs of the same sample – potentially up to 22 if we assume the above analysis is correct – not a particularly realistic number for more sophisticated scientific questions and limited samples or experimental time.

Additionally, the calculation above requires an estimation of the number of degrees of freedom for the t-statistic. Is that the number of droplets, because each droplet can freeze at a different time/temperature, or is that the number of replicates since those are the actual parameters we are comparing? Again, it is unclear.

Of course, the ensemble of droplets is much larger than the sample size of 22 and increasing the number of droplets frozen is usually observed to increase the accuracy of an experiment (and therefore statistical tests based on that experiment). In that case, how would those droplets contribute to the sample size requirement? To our knowledge, there is no quantitative estimate for this effect. So, when asking how many droplets and/or freezing assays are required for a given confidence interval to have a certain margin of error, we have essentially no way of giving a quantitative answer.

This problem has often been ignored in the past (e.g., there is no real mention of how the theory behind confidence intervals works with the sampling methods used in freezing experiments in previous papers we cited in this section). Since this manuscript is offering a new way of doing these statistics that does not have the same problem (although as you rightfully requested, sample size considerations are still important for our technique, albeit quite different), we do not feel that it is within the scope of our work to give quantitative estimates of how inaccurate these techniques are, as it would require building a significant amount of statistical theory for a technique that already has many other issues for this application.

Instead, we have added the following language after the sentence on Z- and t-intervals:

"While it is unclear how many droplets and freezing assays are required for these approximations to be valid under the Central Limit Theorem, in our experience it is unlikely that most existing freezing assay datasets achieve this sample size requirement, since confidence intervals calculated using these techniques often disagree with those calculated using other methods described below and those presented in this study. It is also unclear what exactly a required sample size would mean in this context: the number of droplets is not sufficient, because each droplet does not contribute to every point on the observed ice nucleation spectrum equally. However, the number of separate ice nucleation assays is also not sufficient, as techniques that measure hundreds of droplets in a single assay should require fewer overall assays to calculate accurate statistics because there are more droplets contributing to the accuracy of each point on the measured ice nucleation

[2] Polen, M., Brubaker, T., Somers, J., & Sullivan, R. C. (2018). Cleaning up our water: Reducing interferences from nonhomogeneous freezing of "pure" water in droplet freezing assays of ice-nucleating particles. *Atmospheric Measurement Techniques*, *11*(9), 5315–5334. https://doi.org/10.5194/amt-11-5315-2018

**spectrum. Some combination of the two is required, but there is no existing method by which the accuracy of confidence intervals for an ice nucleation spectrum can be evaluated based on the relevant sample sizes."**

This concept is also discussed in relation to our proposed techniques as follows:

**"Although we cannot theoretically determine the sample sizes required for accurate confidence bands using empirical bootstrapping due to the same limitations discussed previously, the sample sizes required for accurate confidence bands can be empirically evaluated by testing how many assays, droplets, and simulated spectra are required for confidence bands to converge (therefore reducing the uncertainty of the confidence bands due to sample size)."**

p.13, l.40: The chosen sample sizes (e.g., n=91) seem to be arbitrary - please comment.

**The parenthetical statement "(the number of droplets present in the two unaged FUE freezing experiments)" has been added to clarify this.**

p.14, l.56: Replace „per degree Celsius measured" with „per Kelvin".

**Both are valid SI unit choices, and we prefer to use degrees Celsius, in part because this is directly referenced to the melting point of water and thus very relevant for discussing ice nucleation.**

---

## Author Response (AR2)

We thank Referee 2 for their time in reviewing our manuscript again. Responses to each comment along with any changes made are shown below.

1. **In my previous comment, now p. 19, l. 99-00, the authors change is not sufficient. In order to remain balanced, it is important to state that the methods without a background freezing activity instead measure homogeneous ice nucleation. The authors have already cited many homogeneous ice nucleation studies for the droplets-on-substrate approach already in the references section. Please use one of these.**

    This section is only relevant to those studies measuring heterogeneous nucleation, although you are correct that if systems do not have background heterogeneous freezing than the background freezing activity is from homogeneous ice nucleation. This has been clarified as follows:

    > "All droplet-on-substrate methods used to measure heterogeneous IN activity have some level of background freezing activity either from background heterogeneous nucleation or from homogeneous ice nucleation that can change day-to-day depending on the system (Polen et al., 2018; Vali, 2019). For accurate measurements and to compare between instruments, the instrumental background (or homogeneous ice nucleation activity) must be subtracted from any measured heterogeneous IN spectrum."

2. **P. 7, l. 78, please define the FUE acronym. It is first mentioned here without saying what it is referring to.**

    The FUE acronym has now been defined in line 80 as follows:

    > "The Fuego ground PM37 sample (FUE) from Jahn et al. (2019) was tested…"

3. **Section 4.1 and 4.2. The authors review the many ways of calculating confidence intervals around ns, or how to calculate variability from Poisson statistics, etc… However, they do not compare these. The authors make claims that one method may be risky or has false assumptions, however there is no attempt to quantitatively compare these. In other words, Fig 3 only includes their bootstrapping methods, and no other. This must be stated in the manuscript at the end of section 4.2, that it is not the focus of this study to intercompare the various ways error or variability is determined. The quantitative importance of whatever the authors think is risky or assumed on calculated ns variability is not evaluated here, and this should be stated to not mislead readers into thinking one way of doing things is better than the other.**

    While you are correct that we do not quantitatively compare previous methods, a qualitative comparison of the assumptions and approximations made by previous methods is still a valid comparison of sorts. To clarify we have added the following sentence at the end of Section 4.2:

    > "Note it is not the purpose of this study to quantitatively compare methods previously used to calculate uncertainty in IN spectra, and the above discussion is only as a qualitative overview of the assumptions and approximations previous methods use."

4. **P. 11, l. 05. Whenever random sampling is done, the name of the probability distribution from which numbers are sampled must be stated. In this case, it is a discrete uniform distribution function. Please state this.**

    We have specified that the distribution sampled from is a discrete uniform distribution function as requested by adding:

*"…list of freezing temperatures using a discrete uniform distribution function"*

5. **P. 11, l. 06-08. This is identical to discrete inverse transform sampling. This should be stated. In addition to these comments of p. 11, does it matter to your conclusions that the distribution sampled from are discrete? If they were continuous, would anything change? That also may be worth discussing if you have evidence one way or another.**

We do not see why it is relevant to the referenced lines that the underlying algorithm used in the choices function in Python is a form of discrete inverse transform sampling. In our opinion this would be addition of unnecessary jargon, as the algorithm has been specified to sufficient detail in plain English and we have added the specific functional form of the distribution being sampled from as per the previous comment.

6. **Figure 5, caption. There is an extra space.**

Thank you for catching that.

7. **P. 18, l. 69-70, check capitalization**

Changed 'bands, If' to 'bands, 'if'

8. **Figure 7, caption. Check the colors. Purple or pink?**

This spectrum is changed to goldenrod for colorblind friendliness now in the high-resolution figure to match a previous plot, this has been changed in the caption.

Referee 1 made this suggestion:

**There is one part in the introduction (l. 32-41) that should be revised for clarity/readability - the terms "IN activity" and "ice nucleation rate" are used interchangeably or in a circular fashion.**

In that section we do already state that the ice nucleation rate is one way that the IN activity is determined. So this should already be clearly explained.

**Also, "singular-stochastic model" sounds contradictory.**

[revised manuscript text omitted]